# Landscape of epithelial–mesenchymal plasticity as an emergent property of coordinated teams in regulatory networks

**Kishore Hari[1], Varun Ullanat[2], Archana Balasubramanian[3], Aditi Gopalan[2], Mohit Kumar Jolly[1]***

[1]Centre for BioSystems Science and Engineering, Indian Institute of Science Bangalore, Bangalore, India; [2]Department of Biotechnology, RV College of Engineering, Bangalore, India; [3]Department of Biotechnology, PES University, Bangalore, India

**\*For correspondence:**
mkjolly@iisc.ac.in

**Competing interest:** The authors declare that no competing interests exist.

**Abstract** Elucidating the design principles of regulatory networks driving cellular decision-making has fundamental implications in mapping and eventually controlling cell-fate decisions. Despite being complex, these regulatory networks often only give rise to a few phenotypes. Previously, we identified two 'teams' of nodes in a small cell lung cancer regulatory network that constrained the phenotypic repertoire and aligned strongly with the dominant phenotypes obtained from network simulations (Chauhan et al., 2021). However, it remained elusive whether these 'teams' exist in other networks, and how do they shape the phenotypic landscape. Here, we demonstrate that five different networks of varying sizes governing epithelial–mesenchymal plasticity comprised of two 'teams' of players – one comprised of canonical drivers of epithelial phenotype and the other containing the mesenchymal inducers. These 'teams' are specific to the topology of these regulatory networks and orchestrate a bimodal phenotypic landscape with the epithelial and mesenchymal phenotypes being more frequent and dynamically robust to perturbations, relative to the intermediary/hybrid epithelial/mesenchymal ones. Our analysis reveals that network topology alone can contain information about corresponding phenotypic distributions, thus obviating the need to simulate them. We propose 'teams' of nodes as a network design principle that can drive cell-fate canalization in diverse decision-making processes.

## Editor's evaluation

This important article identifies topological metrics in gene regulatory networks that potentially predict the kinds of phenotypic steady states that the network allows. In particular, for epithelial–mesenchymal plasticity, the authors show compellingly that the relevant gene regulatory networks are structured as 'teams' that may be 'strong,' yielding stable phenotypes, or 'weak,' yielding unstable phenotypes prone to plasticity. The work would be of interest to researchers interested in systems biology and the nonlinear dynamics of biological systems, as well as biologists interested in gene regulatory networks and their (mis)functioning in cancer cells.

## Introduction

Understanding the principles of cellular decision-making is a fundamental question in cellular and developmental biology, with implications for mapping and eventually controlling cellular reprogramming

and disease progression (*Balázsi et al., 2011*; *Guantes and Poyatos, 2008*; *Prochazka et al., 2017*). These decisions are often orchestrated through the emergent dynamics of complex regulatory networks operating at multiple levels, including signaling, protein–protein interaction, and transcriptional activation/inhibition. Nonlinear interactions in various such networks can enable emergent dynamics such as multistability and hysteresis (cellular memory) to facilitate adaptation to multiple stresses (*Agozzino et al., 2020*; *Ozbudak et al., 2004*). A better understanding of the underlying dynamics can also accelerate the design of synthetic circuits to achieve specific objectives. Thus, elucidating how specific regulatory networks lead to different emergent dynamics is instrumental for understanding how cells decide among multiple possible fates/phenotypes to choose from and how such transitions can be controlled for directing cellular reprogramming to achieve specific desired scenarios, such as 'differentiation therapy' (*Enane et al., 2018*) for cancers and/or reprogramming pancreatic cells to make insulin (*McKimpson and Accili, 2019*). Questions related to the transition of cells from one state/phenotype to another reversibly or irreversibly during cellular differentiation have been investigated for almost five decades (*Newman, 2020*). We now know that many cellular decisions are often mediated by 'network motifs' such as a toggle switch – a mutually inhibitory feedback loop between the two 'master regulators' of sibling cell fates (*Zhou and Huang, 2011*). Such loops often drive two diverging decision-making trajectories in Waddington's landscape, representing different possible 'terminal' states to which a cell can converge. Thus, a 'toggle switch' between two nodes, A and B, leads to two states – (high A, low B) and (low A, high B) – each representing a different phenotype. However, decision-making may involve much larger regulatory networks, often involving multiple feedback loops. For instance, the global regulatory network in *Escherichia coli* has approximately 150 transcription factors (TFs) (*Fang et al., 2017*). Similarly, networks driving epithelial–mesenchymal plasticity (EMP) in cancer cells can have over 50 players (*Font-Clos et al., 2018*). Despite their complexity, many of these networks robustly lead to only a limited number of phenotypes, a process termed 'canalization' (*Gates et al., 2021*). This observation raises the question of whether these networks constitute topological signatures capable of constraining the corresponding possible phenotypic repertoire. We recently investigated the dynamics of a complex regulatory network (33 nodes, 357 edges) that led to only four phenotypes in small cell lung cancer (SCLC) (*Chauhan et al., 2021*). We demonstrated that this network consisted of two 'teams' of nodes such that members in a team activated each other directly/indirectly, but members across teams inhibited each other. This topological feature reduced this complex network effectively into a 'toggle switch' between teams, thus leading to a small number of phenotypes. However, many questions remain unanswered: (a) Can the presence of 'teams' be witnessed in other regulatory networks? If yes, do they constrain the phenotypic space in those networks too? (b) Do these 'teams' also make these biological networks/ phenotypes robust to various perturbations? (c) Can the team strength be used to predict the frequency of different phenotypes without performing dynamic simulations? Here, we investigate the dynamics and topological hallmarks of five networks of different sizes, all implicated in regulating EMP in diverse biological contexts. First, we established that the topology of these networks could give rise to a largely bimodal phenotypic stability landscape. Second, by analyzing their network topology, we found that all these networks consist of 'teams' of nodes; one of these 'teams' comprised drivers/ stabilizers of mesenchymal phenotype, while the other one has those for the epithelial phenotype. This 'team' structure was largely lost upon disrupting the network topology by shuffling/randomizing edges. Third, our discrete parameter-independent and continuous parameter-agnostic simulations show that these 'teams' are integral to stabilizing epithelial and mesenchymal phenotypes, as demonstrated via various stability metrics. Thus, the hybrid epithelial/mesenchymal phenotypes were less frequent and less resilient to dynamic perturbations. Overall, we show that the strength of 'teams' in a regulatory network directly shapes the emergent largely bimodal phenotypic landscape, thus offering a network topology-based metric to identify phenotypic distributions without performing any simulations. The topological signatures and metrics identified here can also be applied to other cellular decision-making instances to unravel their underlying fundamental dynamic hallmarks.

## Results

## EMP network topology can lead to a largely bimodal phenotypic stability landscape

EMP is a developmental program that enables cells to attain phenotypes in a spectrum ranging from epithelial (E) to hybrid (E/M) to mesenchymal (M) phenotypes. While identifying the number of possible hybrid phenotypes is an ongoing research area (*Brown et al., 2022*; *Cook and Vanderhyden, 2020*; *Karacosta et al., 2019*), their stability characteristics have been well-studied. The epithelial and mesenchymal phenotypes at the end of the spectrum (referred to as terminal phenotypes here) have been observed to be more 'stable' than the hybrid phenotypes in various contexts. These stability differences lead to an uneven, bimodal phenotypic stability 'landscape', that is, a highly stable group of terminal phenotypes and a weakly stable group of hybrid phenotypes (*Pastushenko et al., 2018*). To understand the role of network topology in the emergence of such a bimodal landscape, we chose

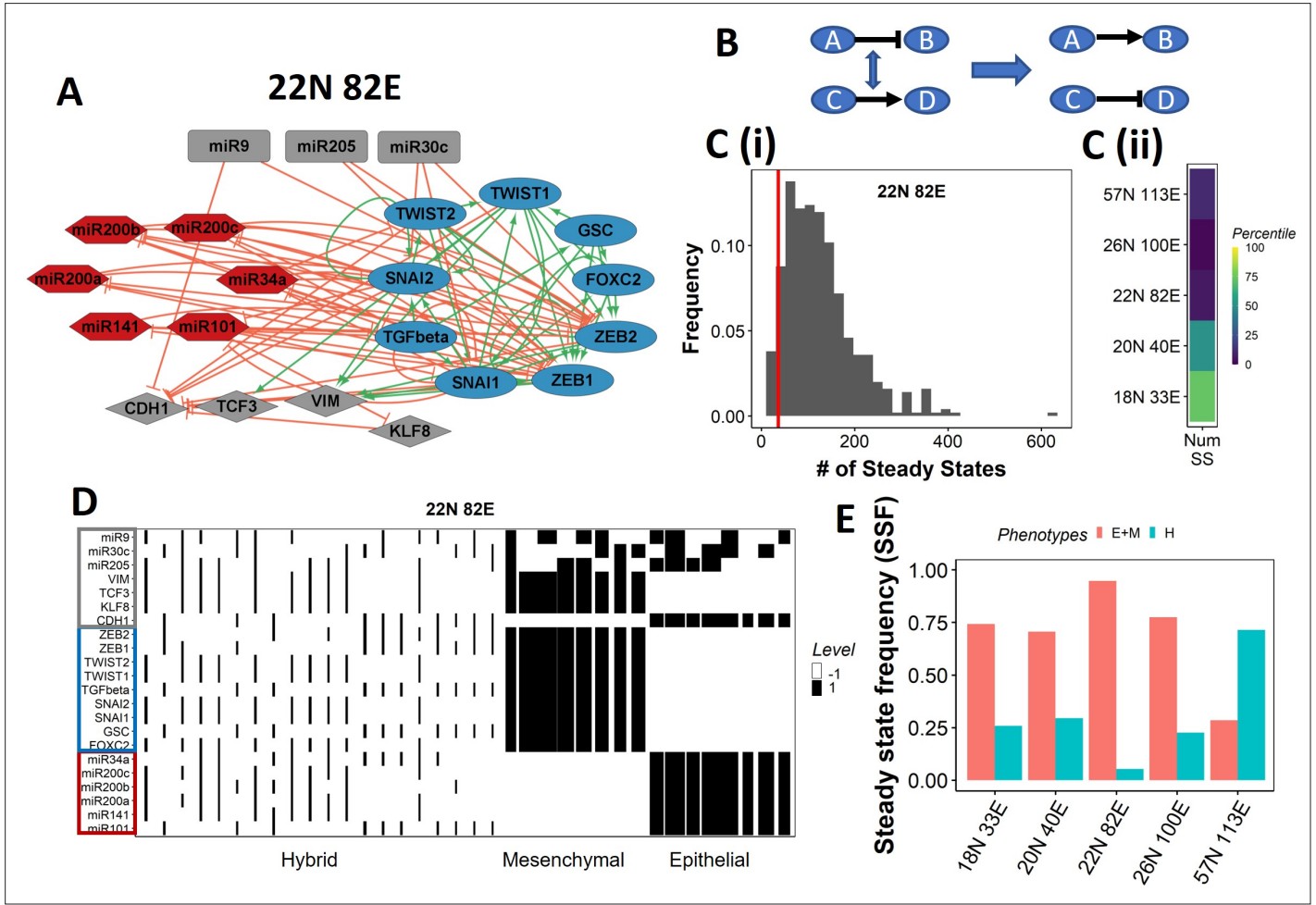

**Figure 1.** Epithelial–mesenchymal plasticity (EMP) network topology can result in a bimodal phenotypic stability landscape. (**A**) EMP network of size 22N 82E, where N stands for number of nodes and E stands for number of edges. (**B**) Demonstration of network randomization. (**C**) (**i**) Distribution of number of steady states in random networks of size 22N 82E. The wild-type EMP network of the same size is represented using the red line. (**ii**) Percentile of the WT network in the distribution of the number of steady states in random networks. (**D**) Heatmap depicting the steady states of the 22N 82E network. Each column represents a steady state. Each row represents a node. White cells indicate low-expressing/inactive node (–1) in a state and black cells indicate high expression/active (1). The width of each column is proportional to the frequency of the given steady state. (**E**) Comparison of the cumulative frequency of the terminal (epithelial and mesenchymal) states vs. that of the hybrid states for all five EMP networks.

The online version of this article includes the following figure supplement(s) for figure 1:

**Figure supplement 1.** EMP networks and their steady state patterns.

**Figure supplement 2.** Heatmaps depicting the steady states of the epithelial–mesenchymal plasticity (EMP) networks.

to investigate a collection of five EMP networks that have been shown to be previously investigated via different simulation formalisms (*Silveira and Mombach, 2020*; *Silveira et al., 2020*; *Huang et al., 2017*; *Tripathi et al., 2020a*; *Font-Clos et al., 2018*; *Figure 1A*, *Figure 1—figure supplement 1A*). We chose these networks to range over various sizes and densities (18N 33E to 57N 113E, where N is the number of nodes and E is the number of edges in the network). Each of these networks depicts the regulation of EMP at the transcriptional/post-transcriptional level (compiled under different biological/experimental contexts). Therefore, each node is either a TF or a micro-RNA, and each edge represents transcriptional or post-transcriptional activation or inhibition. We classify these nodes into two categories based on their topological configuration: 'peripheral' nodes and 'core' nodes. The peripheral nodes are the ones that either have no incoming edges (i.e., input nodes/signals) or no outgoing edges (i.e., output nodes). Based on their biochemical nature, we further classify the core nodes as epithelial nodes, which are the drivers of the epithelial phenotype, and mesenchymal nodes, known to be the drivers of the mesenchymal phenotype. In the 22N 82E network shown in *Figure 1A*, the mesenchymal nodes are TWIST1/2, GSC, FOXC2, ZEB1/2, SNAI1/2, and TGFb; the epithelial nodes are miR-200a, miR-200b, miR-200c, miR-34a, miR-141, and miR-101; and the peripheral nodes are KLF8, TCF3, VIM, CDH1, miR205, miR30c, and miR9 (*Tripathi et al., 2020b*; *Deshmukh et al., 2021*; *Brabletz and Brabletz, 2010*). Similar classification of nodes has been implemented for other networks (*Figure 1—figure supplement 1A*; *Silveira and Mombach, 2020*; *Silveira et al., 2020*; *Huang et al., 2017*; *Tripathi et al., 2020a*; *Font-Clos et al., 2018*). The interactions between these nodes are referred to as edges and are either activating or inhibiting in nature. We simulate the dynamics of these networks using a threshold-based, parameter-independent, Boolean formalism (*Font-Clos et al., 2018*), where each node can be either active (1) or inactive (–1). We define the state of a node as an array of –1s and 1s, where each element of the array depicts the activity of a node. The activity of each node is affected by the activity of all the incoming edges based on a majority rule, that is, if there are more inhibiting edges active, the node gets inactivated and vice versa (see 'Methods'). We update the state of the network using an asynchronous formalism where only one node (randomly chosen) is updated at a time step. This formalism captures the inherent stochasticity in the emergent dynamics of these networks. We simulated these networks until the system reaches a steady state, that is, the state of the network does not change with time.

Despite the size and complexity of these networks, we noticed a relatively smaller number of states, indicating canalization (36 steady states for the 22N 82E network, out of the $2^{22}$ possible states, *Figure 1—figure supplement 1B*). To check whether this property depends on the network topology, we generated 500 random (hypothetical) networks by randomly selecting and swapping different pairs of edges in the network. This ensured that the in and out degrees of all nodes remain the same, but the way they are connected (network topology) changes (*Figure 1B*, 'Methods'). The wild-type (WT) EMP networks showed a much smaller number of steady states than those shown by most random networks (*Figure 1C*), suggesting that the topology of the EMP networks plays a significant role in limiting the phenotypic repertoire.

We represent the steady states of WT EMP networks in a heatmap, where each row corresponds to one node in the network, and each column corresponds to a steady state (*Figure 1D*, *Figure 1—figure supplement 2*). As expected, we see three categories of states: epithelial states that have all epithelial nodes (highlighted by a blue border) as active (dark cells) and all mesenchymal nodes (highlighted by a red border) as inactive (white cells); mesenchymal states that have all mesenchymal nodes as active and epithelial nodes as inactive; and hybrid states that have one of the possible combinations of epithelial and mesenchymal nodes as active. The steady-state frequency (SSF) is calculated as the fraction of initial conditions that converge to the given steady state. We represent the SSF for each steady state as the width of the corresponding column. The epithelial and mesenchymal states account for >70% of the SSF in four of the five EMP networks (*Figure 1E*). These results indicate the emergence of the experimentally observed uneven (bimodal) stability landscape (*Pastushenko et al., 2018*) that can be explained from the network topology alone, without any specific kinetic parameters.

## Bimodality of the phenotypic landscape is weakened upon randomizing the network topology

SSF, the fraction of the possible states that converge to a given steady state, is a measure of the stability of a steady state. SSF shows a bimodal distribution in WT EMP networks (*Figure 2A*), following

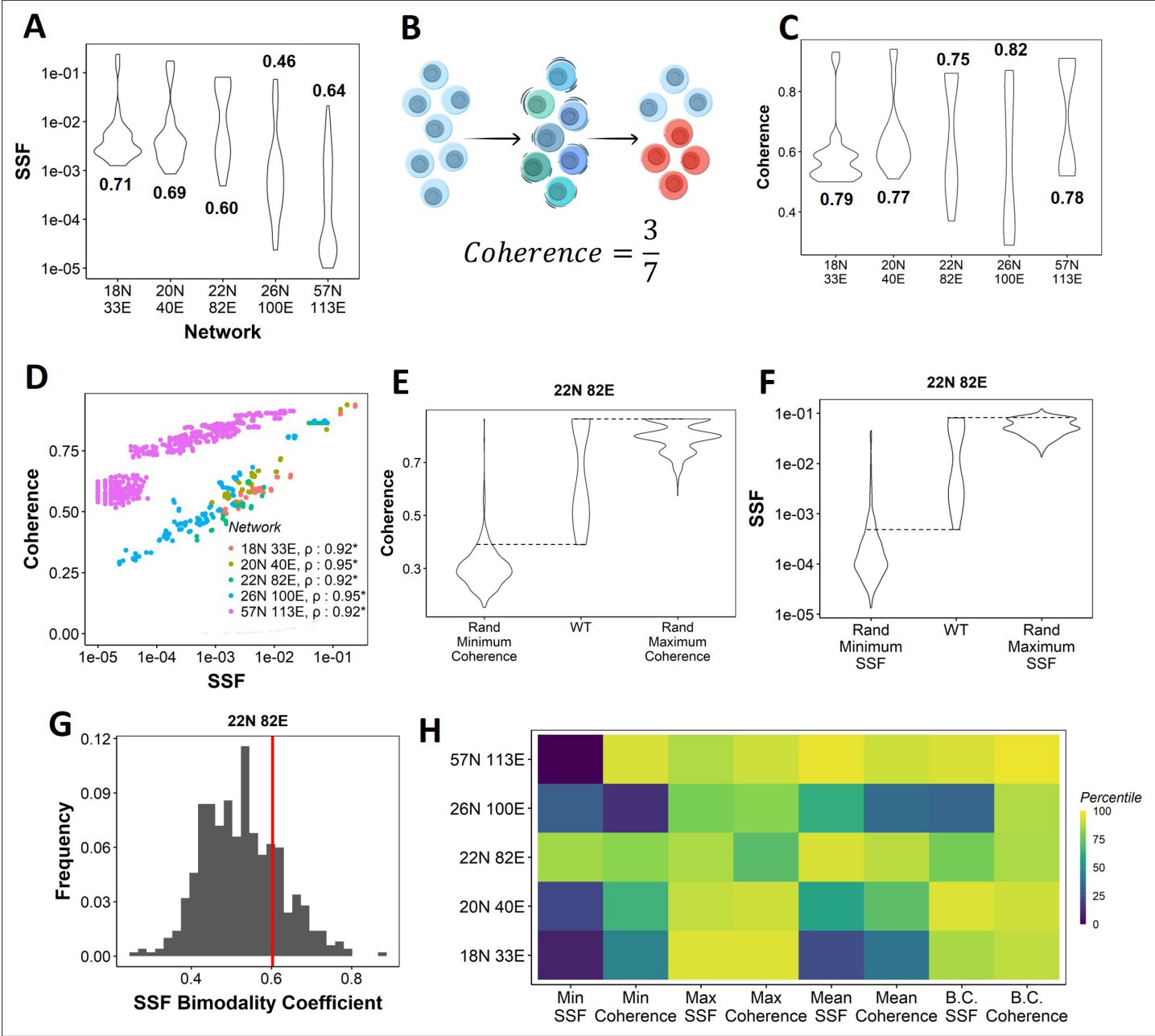

**Figure 2.** Dynamic traits of phenotypes observed from wild-type (WT) epithelial–mesenchymal plasticity (EMP) networks and their randomized counterparts. (**A**) Distribution of steady-state frequency (SSF) of the steady states obtained for the five EMP networks, in log10 scale. The corresponding Sarle's bimodality coefficients have been reported. A value greater than 0.55 indicates bimodality. (**B**) Depiction of coherence calculation. The blue balls indicate unperturbed steady state (P1, say). The green and dark blue balls represent the perturbations given to the steady state. The red balls represent a different steady state that the system reached after the perturbation. The fraction of perturbations that reverted to the original state P1 (3 out of 7 balls) is calculated as coherence. (**C**) Similar to (**A**), but for coherence of the steady states of WT EMP networks. (**D**) Scatterplot between coherence and SSF of WT EMP networks. Spearman's correlation coefficient for each network has been reported. *p<0.05. (**E**) Comparison of the distribution of coherence of the steady states of WT EMP network (22N 82E), with the distribution of maximum coherence values and minimum coherence values of the corresponding random networks. (**F**) Similar to (**E**), but for SSF. (**G**) Distribution of the SSF bimodality coefficients for random networks of size 22N 82E. The red vertical line represents the WT network. (**H**) Percentile of the WT networks in the distribution of multiple stability metrics obtained from random networks.

The online version of this article includes the following figure supplement(s) for figure 2:

**Figure supplement 1.** Coherence and steady-state frequencies for EMP networks.

the observation that terminal phenotypes have much higher SSF than hybrid phenotypes (*Figure 1D and E*, *Figure 1—figure supplement 2*). SSF measures the global stability of a given steady state/phenotype in the state space. Specifically, SSF is an estimate of the fraction of the state space that can converge to the given steady state. Similarly, the local stability of the steady state can be estimated by measuring the fraction of neighboring states in the state-space that converges to the steady state. We hence used a metric – 'coherence' - based on the idea of local stability proposed previously (*Willadsen and Wiles, 2007*). The calculation of coherence follows the perturbation procedure depicted in *Figure 2B*. For each steady state, we perturb the activity of one node at a time (active nodes flipped to be inactive and vice versa) and simulate the dynamics till a steady state is reached. We then measure the fraction of times the original steady state is regained upon such perturbations and label it as the coherence of the corresponding steady state ('Methods').

Next, we quantified the SSF and coherence for all steady states from the randomized (hypothetical) networks and compared the stability of the phenotypes emerging from WT and random networks. We calculated the minimum, maximum, and mean coherence and SSF across all steady states for a given random network and plotted the distribution of these values (*Figure 2E and F*). When comparing with the corresponding values observed for the WT networks, we found that the maximum coherence seen for the 22N 82E WT network was more than that seen for most of the corresponding random networks (compare columns 2 and 3 in *Figure 2E*); consistent results were obtained for other WT networks (*Figure 2—figure supplement 1A*).

Like SSF, coherence of the steady states of EMP networks also shows a bimodal distribution, endorsing the bimodality in the stability landscape of EMP phenotypes (*Figure 2C*). We further observed a strong positive correlation between coherence and SSF for all five WT EMP networks (*Figure 2D*). In WT networks, we expect terminal phenotypes to have a higher coherence based on the strong positive correlation between SSF and coherence (*Figure 2D*). Conversely, we expect hybrid phenotypes to have a reduced coherence. As a metric, coherence provides the following advantages over SSF: (1) coherence is a perturbation-based measure and therefore provides a dynamic perspective of the stability of the steady states. In EMP, coherence can be visualized as the effect of a weak EMT (Epithelial to Mesenchymal Transition)/MET (Mesenchymal to Epithelial Transition) -inducing signal. (2) Coherence being a local stability measure is less dependent on the other steady states of the network. Therefore, the absolute coherence values can be compared across networks, unlike SSF (compare the range of y-axis values in *Figure 2A* for SSF vs. *Figure 2C* for coherence).

When comparing the patterns for minimum coherence, we observed similar trends for the 22N 82E network (*Figure 2E*, columns 1 and 2) but not for 3 of the four remaining WT EMP networks (*Figure 2—figure supplement 1A*). Maximum and minimum SSF behave similarly to the coherence, that is, the WT maximum SSF is higher than most random networks. In contrast, minimum SSF is not consistent (*Figure 2F*, *Figure 2—figure supplement 1B*, columns 1–3). Furthermore, we compared the bimodality coefficient of SSF of the WT 22N 82E network against the distribution obtained from random networks and found it to be higher than most random networks (*Figure 2G*). The trend holds for other EMP networks and the bimodality coefficient of coherence (*Figure 2—figure supplement 1C and D*). To quantify these trends better, we obtained percentiles for the WT values of all eight (four for coherence, four for SSF) of these metrics in the corresponding random network distributions (*Figure 2H*). The coherence bimodality coefficient of all five WT EMP networks is greater than 80% of the corresponding random networks. Similarly, we find the SSF bimodality coefficient to be higher than 75% of the random networks in all cases except the networks of size 26N 100E. Furthermore, we find that the maximum coherence and maximum SSF for all five WT EMP networks are higher than at least 75% of the corresponding random networks. Such a trend was not consistently seen for minimum and mean coherence and SSF values (*Figure 2H*). In WT EMP networks, the maximum coherence and SSF represent the terminal phenotypes, and minimum coherence and SSF represent the hybrid phenotypes. Hence, these results suggest that the WT networks show a more robust control in maintaining the high stability of terminal phenotypes but exhibit a weaker control over the stability of hybrid phenotypes. Hence, we hypothesize that the bimodal landscape observed in WT EMP networks is an emergent feature of their network topology.

We then investigated what factors determine the emergence of the bimodal landscape, where terminal phenotypes show higher relative stability (SSF and coherence) than hybrid phenotypes. One possible way to stabilize a state is to have a strong agreement between the state configuration and

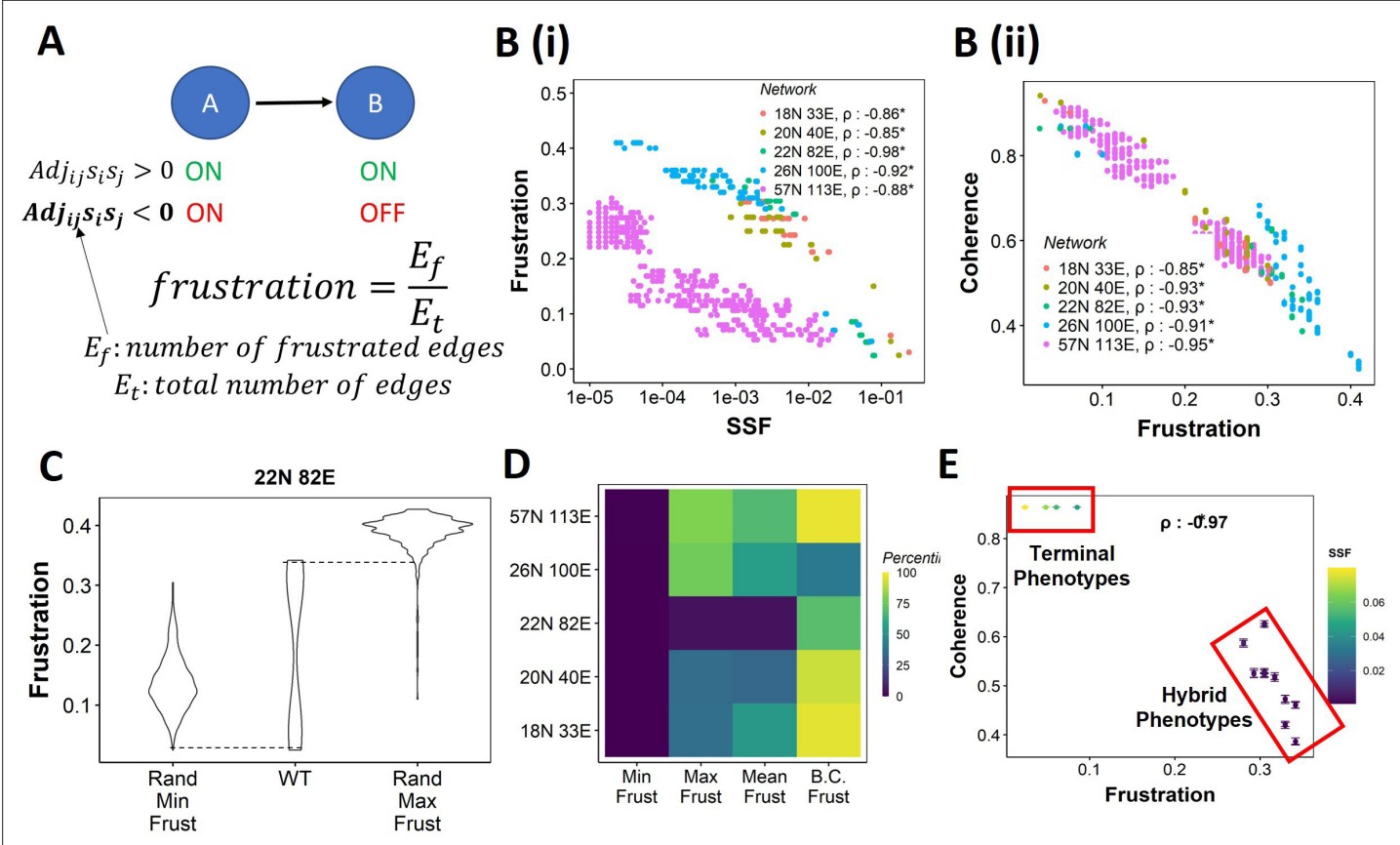

**Figure 3.** Frustration as a stability metric for wild-type (WT) and random networks. (**A**) Depiction of calculation of frustration. represents the interaction from $i$th node to $j$th node, $s_i$ and $s_j$ represent the activity of the $i$th node and the $j$th node for a given state $S$. (**B**) Scatterplot between frustration and (**i**) steady-state frequency (SSF), (**ii**) coherence for WT epithelial–mesenchymal plasticity (EMP) networks. Spearman's correlation coefficient for each network has been reported. *p<0.05. Each dot corresponds to a steady state for the given network. (**C**) Comparison of the distribution of frustration of the steady states of WT EMP network (22N 82E), with the distribution of maximum frustration values and minimum frustration values of the corresponding random networks. (**D**) Heatmap of the percentile of WT network values in the random network value distribution for the minimum, maximum, and mean frustration. (**E**) Representation of WT steady states in a scatterplot of frustration and coherence with the color representing the corresponding SSF. Spearman's correlation coefficient between the axis metrics is reported. *p<0.05. The terminal and hybrid phenotypes are highlighted with red rectangles.

The online version of this article includes the following figure supplement(s) for figure 3:

**Figure supplement 1.** Association among SSF, coherence and frustration across EMP networks.

the network topology. For instance, if A activates B in a network, and for a given state, if A and B have opposite values (e.g., –1,1), there is a disagreement between the state configuration and the particular edge. Such disagreement is referred to as the edge of being frustrated. For a given network and a state, we measure the fraction of such frustrated edges and call this fraction the frustration of the state (**Figure 3A**, **Tripathi et al., 2020a**). The higher the frustration, the lesser the chance of a state being stable (see Appendix 1, Appendix text).

We quantified the frustration for each state for each EMP network and observed a strong negative correlation of frustration with both SSF and coherence (**Figure 3B and C**). We then calculated the pairwise correlation for the three metrics in random networks. While the correlation between SSF and coherence was consistently positive (**Figure 3—figure supplement 1A**), the correlation of frustration with SSF and coherence was positive in some random networks (**Figure 3—figure supplement 1B and C**, 18N, and 20N networks). These results suggest that, while frustration, a measure of the support that the network topology provides to a given state, can explain the stability of a state in terms of SSF and coherence in WT networks, the same may not be accurate for random networks.

Next, we compared the minimum and maximum frustration values for random networks with those of corresponding WT EMP networks. As noted earlier for coherence and SSF (*Figure 2E and F*, *Figure 2—figure supplement 1A and B*), not all metrics (maximum, minimum, mean) of frustration show the same trend across EMP networks. While minimum frustration of EMP networks was lower than that of most random networks, the maximum and mean frustration do not necessarily follow the same trend (*Figure 3D*, *Figure 3—figure supplement 1D*). Furthermore, the bimodality coefficient of frustration for WT networks is also higher than most of the random networks (*Figure 3D*), similar to that observed in SSF and coherence (*Figure 2G and H*, *Figure 2—figure supplement 1C and D*). Finally, having investigated three different metrics – coherence, SSF, and frustration – to measure state stability, we collated this information for the steady states of WT networks on a scatterplot of coherence vs. frustration, with the color marked by SSF. In this plot, we can clearly visualize the bimodal landscape, that is, the terminal phenotypes have high SSF, high coherence, and low frustration, and hybrid phenotypes, on the other hand, have low SSF, low coherence, and high frustration (*Figure 3F*).

## EMP networks contain two mutually inhibiting teams of nodes

Next, we asked what salient features of network topology for WT EMP networks may underlie their specific bimodal phenotypic stability landscape and the limited phenotypic repertoire. We had earlier observed that a large and complex network regulating phenotypic heterogeneity in SCLC gave rise to predominantly only four phenotypes and, consequently, a bimodal phenotypic stability landscape (*Chauhan et al., 2021*). The hallmark of the topology of the SCLC network was the presence of 'teams' of nodes mutually inhibiting each other. Furthermore, the composition of the dominant phenotypes perfectly coincided with the composition of the 'teams.' Hence, we hypothesized that these EMP networks consist of similar 'teams' of nodes and that these teams underlie their bimodal stability landscape.

Unlike the SCLC network, these EMP networks were highly sparse, that is, the ratio between the number of edges ($E$) and the number of possible edges ($N^2$) given the number of nodes ($N$) is very less (5–15%) (*Figure 4A*). However, pairs of nodes can influence each other not only directly but also via indirect paths (connected edges) of length >1 that can connect them. Thus, we decided to use the pairwise influence among the nodes of these networks rather than just the direct interactions among them to analyze the structure of these networks. This 'influence matrix' represents the effective regulation of one node by another when many different indirect paths are also counted (up to path length <=10) in addition to direct regulation. Each path is assigned a weight inversely proportional to its length while calculating the influence matrix (see the formula below *Figure 4A* and Methods section : *Figure 4*). To check whether the influence matrix can explain phenotypic stability, we calculated the 'strength' of each steady state corresponding to the influence matrix (Appendix 1 'Methods'). While in WT networks the state strength was higher for terminal phenotypes, the same was not true in random networks (*Figure 4—figure supplement 1A–C*). Furthermore, the correlation between state strength and stability metrics was weak in random networks as compared to that of WT networks (*Figure 4—figure supplement 1D*), suggesting that the influence matrix alone is not enough to explain phenotypic stability.

To investigate the relationship between team structure and stability of the phenotypes, we defined a metric called team strength that quantifies the strength of teams in a given network (formula below *Figure 4B*). We first identified teams in random networks via hierarchical clustering of the corresponding influence matrix and calculated the corresponding team strength. We find that the WT EMP networks have stronger teams than most (>98%) of their randomized counterparts (*Figure 4C*, *Figure 4—figure supplement 2B*), indicating that the team structure is a unique topological feature of the WT EMP networks.

We find that the influence matrix, when hierarchically clustered, can be divided into two teams of core nodes (*Figure 4B*, *Figure 4—figure supplement 2A*), similar to SCLC (*Chauhan et al., 2021*). A team here is defined as a collection of nodes that influence each other positively, and members belonging to different teams negatively influence one another. Furthermore, the two teams in WT EMP networks also had distinct biochemical characteristics. In the influence matrix depicted here, team 1 consists of mesenchymal core nodes (blue rectangle in *Figure 4B*, *Figure 4—figure supplement 2A*), and team 2 consists of core epithelial nodes (red rectangle). These teams collectively have a negative influence on each other. Hence, the structure of the influence matrix considering only core

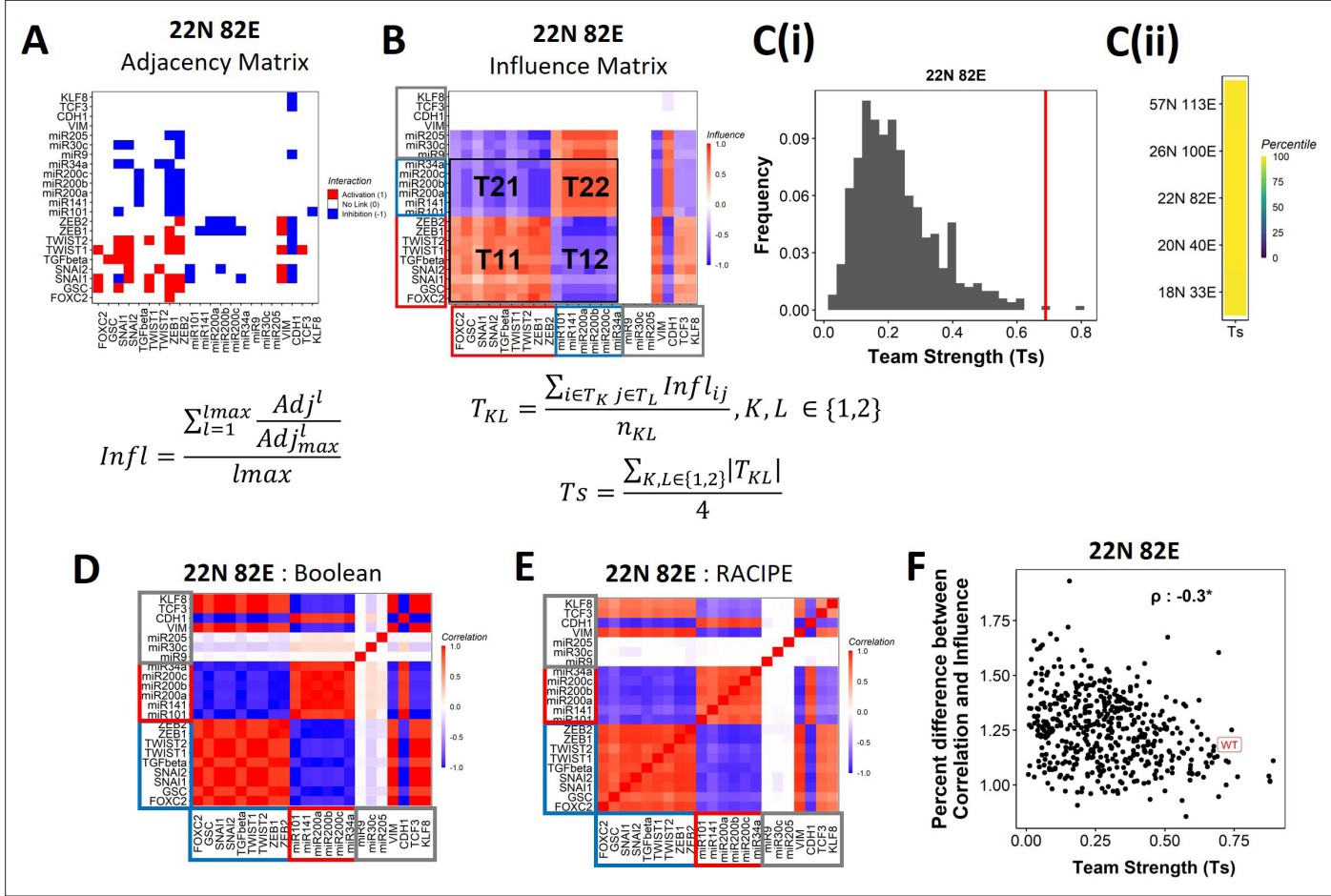

$$Infl = \frac{\sum_{l=1}^{lmax} \frac{Adj^l}{Adj^l_{max}}}{lmax}$$

$$T_{KL} = \frac{\sum_{i \in T_K \, j \in T_L} Infl_{ij}}{n_{KL}}, K, L \in \{1,2\}$$

$$Ts = \frac{\sum_{K,L \in \{1,2\}} |T_{KL}|}{4}$$

**Figure 4.** Epithelial–mesenchymal plasticity (EMP) networks consist of two well-coordinated teams. (**A**) Adjacency matrix of the 22N 82E network. Each row depicts the links originating from the node (i.e., input) corresponding to the row (y-axis) and all other nodes (x-axis, outputs). The color represents the nature of the edge: red for activating links, blue for inhibiting links, and white for no links. The formula for the conversion of adjacency matrix to influence matrix is given below the panel, where $Adj$ is the adjacency matrix, $Adj_{max}$ is the adjacency matrix with all –1s replaced with 1s. $Adj^l$ is the adjacency matrix raised to the power of $l$. The division $\frac{Adj^l}{Adj^l_{max}}$ is element-wise. $lmax$ is the maximum path length considered for calculating the influence. (**B**) The influence matrix for the 22N 82E network. The signal and output nodes (peripheral) are highlighted with a gray box, the mesenchymal nodes (team 1) are highlighted with a blue box, and the epithelial nodes (team 2) are highlighted with a red box. The formula for team strength (Ts) is given below the influence matrix. $T_1$ and $T_2$ represent the two teams of nodes in the network (epithelial and mesenchymal nodes, respectively). $n_{KL}$ is the number of cells in the rectangle $T_{KL}$ (**C**) (**i**) Distribution of team strength (Ts) for random networks of size 22N 82E. The Ts value for the corresponding wild-type (WT) EMP network has been highlighted using the red vertical line. (**ii**) Percentiles corresponding to the WT team strength in the corresponding distribution obtained from random networks for networks of all sizes (y-axis). (**D**) Correlation matrix for the expression levels of nodes of the 22N 82E network, as obtained by the Boolean formalism. (**E**) Same as (**D**) but for RAndom CIrcuit PErturbaiton (RACIPE). (**F**) Scatterplot of the difference between the influence matrix and Boolean correlation matrix (y-axis) and the mean group strength of the network (x-axis) for random networks of size 22N 82E. The wild-type EMP network is highlighted in red. Spearman's correlation coefficient is reported. *p<0.05.

The online version of this article includes the following figure supplement(s) for figure 4:

**Figure supplement 1.** Influence matrices and correlation matrices for EMP networks.

**Figure supplement 2.** Steady-state frequency and state strength calculations for EMP networks.

nodes resembles a toggle switch with self-activation formed by the two teams, with each team operating as a single entity. Furthermore, the composition of teams coincides with the composition of the terminal phenotypes (epithelial and mesenchymal phenotypes, compare *Figure 1D* and *Figure 4B*), leading to the hypothesis that the team structure can contribute to a bimodal EMP landscape by stabilizing the terminal phenotypes.

We further explored the connection between teams of nodes in the influence matrix and the phenotypic landscape by calculating the pairwise correlation between the steady states of all pairs of nodes

from Boolean simulations across multiple random initial conditions ('Methods'). We performed hierarchical clustering on the correlation matrix thus obtained. The resultant matrix bore striking visual similarity to the influence matrix and showed the same two teams of nodes as that seen in the corresponding influence matrix (compare *Figure 4D* with *Figure 4B*; *Figure 4—figure supplement 2C* with *Figure 4—figure supplement 2A*). Interestingly, the simulations of the network using an ODE-based, parameter agnostic method called RAndom CIrcuit PErturbaiton (RACIPE) (*Huang et al., 2017*, Appendix 1 'Methods') also yielded a very similar correlation matrix (*Figure 4E*, *Figure 4—figure supplement 2D*), suggesting that the 'teams' identified in influence matrix (without performing any dynamic simulations) are conserved in the corresponding correlation matrix (identified after simulations).

We quantified the difference between the influence matrix and Boolean correlation matrix for the biological network (WT case) as well as the hypothetical networks (see 'Methods'). Intriguingly, the difference between the matrices was lower for the biological networks as compared to that of the hypothetical networks. Also, the hypothetical networks that have a relatively higher team strength showed a lower difference between influence and correlation matrices, with an overall negative correlation between the difference and the team strength (*Figure 4F*, *Figure 4—figure supplement 2E*). These results suggest a possible causative relationship between the existence of the team structure of nodes and the emergent dynamic phenotypes of a network. Furthermore, for all networks taken together (WT and random), the correlation matrix and influence matrix differ by a maximum of 2–3%, suggesting that the influence matrix can be a good predictor of the correlation matrix, irrespective of the strength of teams observed in a given network.

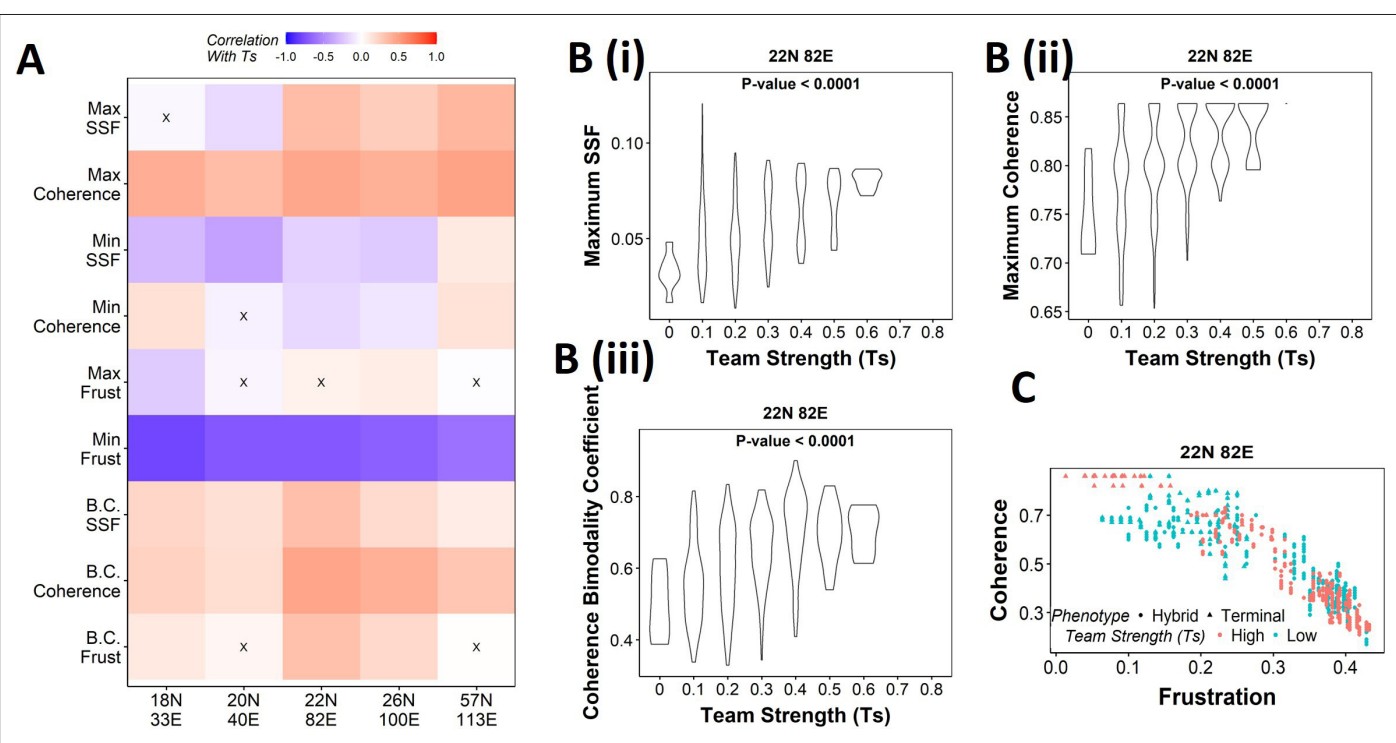

**Figure 5.** Strong teams support the bimodal epithelial–mesenchymal plasticity (EMP) landscape. (**A**) Heatmap depicting the Spearman's correlation of Ts with stability metrics and frustration metrics for random networks of all sizes (y-axis). Insignificant correlations (p>0.05) are marked by 'X.' (**B**) Violin plots depicting the effect of change in Ts against the maximum stability metrics. (**i**) Maximum steady-state frequency (SSF), (**ii**) maximum coherence, and (**iii**) coherence bimodality coefficient for random networks of size 22N 82E. The p-value for one-way ANOVA is reported. (**C**) Scatterplot showing the states of top 10 and bottom 10 (based on mean group strength) random networks of size 22N 82E.

The online version of this article includes the following figure supplement(s) for figure 5:

**Figure supplement 1.** Violin plots depicting the effect of change in Team Strength (Ts) against the maximum stability metrics.

**Figure supplement 2.** Violin plots depicting the effect of change in team strength (Ts) against the minimum stability metrics.

## Strong teams of nodes stabilize terminal phenotypes in WT EMP networks

Our results demonstrate that the WT EMP networks show higher maximum global (measured by SSF) and local (measured by coherence) stability than most of their randomized counterparts (*Figure 2H*), quantities that correspond to the terminal phenotypes in WT EMP networks. However, the stability of phenotypes could not robustly be explained just by considering either the interaction matrix alone (frustration, *Figure 3—figure supplement 1B and C*) or the influence matrix alone (state strength, *Figure 4—figure supplement 1*) for random networks. Additionally, random networks showed weaker teams than the WT EMP networks. Together, these observations strengthen our hypothesis that teams of nodes lead to the stable terminal phenotypes observed in WT EMP networks. To quantify the relationship between team strength and phenotypic stability, we obtained the correlation of the stability metrics (SSF, coherence, and frustration) against the team strength of a given network (*Figure 5A*). Across all five EMP networks and their corresponding random counterparts, Ts correlate consistently positively with maximum coherence and negatively correlated with minimum frustration, suggesting that an increase in team strength increases the stability of the most dominant state emergent from the network.

We then generated violin plots between the stability metrics and team strength to understand the effect of teams on phenotypic stability better. We found that networks with high team strength consistently had high values of maximum SSF and maximum coherence. As the team strength decreases, the maximum stability shows a distribution ranging from high values closer to the high team strength networks to a significantly lower value (*Figure 5Bi,ii*, *Figure 5—figure supplement 1A and B*). The effect of team strength on the minimum stability metrics, however, was negligible, with no trend to observe in the scatterplots either (*Figure 5—figure supplement 2A and B*).

While maximum and minimum stability metrics correspond only to a single emergent phenotype of a network, the bimodality coefficient in the stability metrics can serve well in quantifying the bimodality of the phenotypic stability landscape. Hence, we generated similar plots as above for bimodality coefficients of the stability metrics (*Figure 5Biii*, , *Figure 5—figure supplement 1C*, *Figure 5—figure supplement 2C*). Similar to the maximum stability metrics, we see a positive correlation between the bimodality coefficient and team strength (*Figure 5A*). Furthermore, at high team strength, the emergent phenotypic stability landscape is strongly bimodal, whereas, at low team strength, the networks are not necessarily bimodal.

To better visualize the effect of bimodality coefficient, we took 10 random networks each with the highest and lowest team strengths, and mapped the frustration and coherence of their steady states (*Figure 5C*, *Figure 5—figure supplement 1D*). For networks with high Ts (red points), we clearly see two groups of steady states based on the relative stability (high coherence – low frustration and low coherence – high frustration). While such distinction of two groups of steady states is lost in random networks of low Ts corresponding to 22N 82E, 18N 33E, and 20N 40E, it was maintained in 26N 100E and 57N 113E random networks. This observation strengthens the trend that high team strength corresponds to a bimodal landscape, while at low team strength, bimodality of the phenotypic stability landscape remains unpredictable. Furthermore, strong teams improve the correlation between stability metrics (SSF and coherence) and frustration (*Figure 5—figure supplement 2D and E*), suggesting that the relationship between network topology and state stability is strengthened as the strength of teams increases.

Together, these results suggest that as team strength increases the stability of the most dominant states increases, thereby increasing the bimodality in the phenotypic stability landscape. Additionally, teams increase the agreement between the compositions of steady states and the network topology. However, as teams weaken, the trends do not hold in any particular direction. Hence, we can conclude from these results that teams are sufficient to maximize the phenotypic stability and the bimodality of the landscape of a network but might not be necessary.

## Teams' structure imparts unique transition characteristics to hybrid phenotypes

Having identified that the presence of 'teams' of nodes that cooperate with/activate each other can reinforce a given steady state when a perturbation to one of the node values was made (coherence), we next asked whether the positive reinforcement in a network offered by 'teams' can be extended

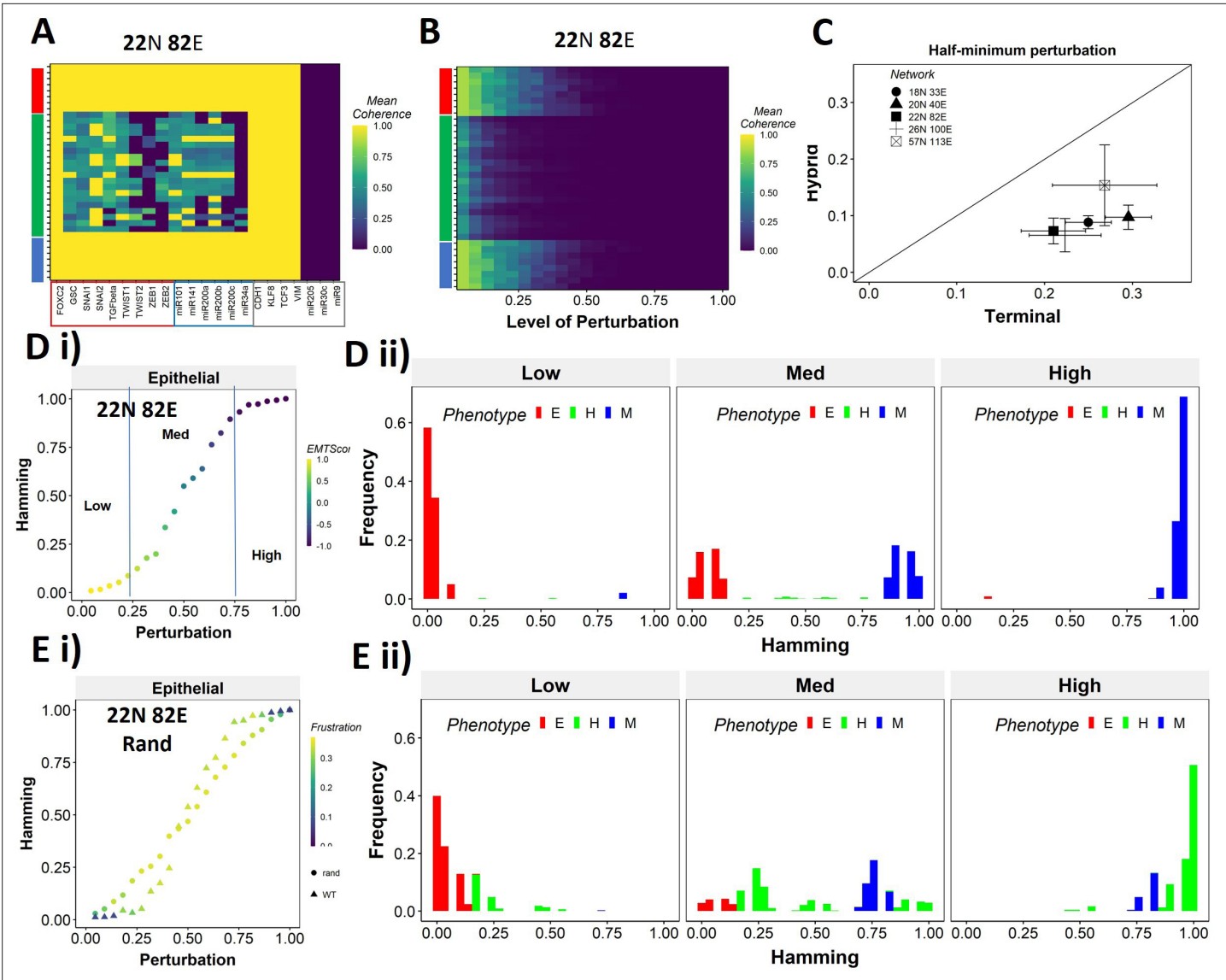

**Figure 6.** Teams of nodes impart distinct dynamic properties to terminal and hybrid phenotypes. (**A**) Heatmap depicting the mean coherence of the Boolean states of 22N 82E wild-type (WT) epithelial–mesenchymal plasticity (EMP) network when each node is individually perturbed. (**B**) Mean coherence of states of 22N 82E EMP network with multiple nodes perturbed at once (Level of perturbation). (**C**) The extent of perturbation required to bring the coherence of terminal phenotypes (x-axis) and hybrid phenotypes (y-axis) below 0.5. x = y line is shown. (**D**) (**i**) Representative mean Hamming distance plot of an epithelial state obtained from 22N 82E WT network. Three levels of perturbation are highlighted based on regions of the sigmoidal plot. (**ii**) Distribution of the Hamming distance from the starting state in (**D**) (**i**) at different levels of perturbation, colored by the phenotype. (**E**) (**i**) Representative mean Hamming distance plot comparing the dynamic transition of an epithelial state from WT and that from a random network. (**ii**) Corresponding distribution of Hamming distances for the random network.

The online version of this article includes the following figure supplement(s) for figure 6:

**Figure supplement 1.** Single and multinode perturbation of WT and random EMP networks.

**Figure supplement 2.** Coherence of calculated values for the core nodes upon perturbing signal nodes in wild-type (WT) epithelial–mesenchymal plasticity (EMP) networks.

to a scenario when more than one node is perturbed. First, we took a closer look at the coherence metric by analyzing the coherence patterns of the steady states when each node of the network is perturbed one at a time (***Figure 6A***, ***Figure 6—figure supplement 1A***). As expected, the terminal phenotypes remained unchanged (coherence = 1) when any one core node (e.g., ZEB1, miR200c, miR34a in ***Figure 6A***) was perturbed, but the hybrid phenotypes had a relatively smaller coherence for such cases. While perturbing signal nodes (miR9, miR30c, miR205 in ***Figure 6A***) results in a coherence

of 0 for all states, it is because signal nodes do not have any inputs and hence cannot revert upon perturbation. However, the configuration of core nodes does not change in terminal phenotypes in such cases (see Methods 4.7.1). Similarly, output nodes (CDH1, VIM, TCF3, KLF8) are always restored back upon perturbation without any effect on the steady state as they do not influence any other node in the network. Next, we perturbed multiple nodes (randomly chosen) simultaneously and calculated the coherence of the steady states. While the terminal states show resilience up to 25% of nodes being perturbed, the hybrid states lose their identity (i.e., switch to another state) upon relatively minor perturbations (*Figure 6B*, *Figure 6—figure supplement 1B*).

We quantified this difference in the EMP networks by measuring the extent of perturbation at which the coherence goes below 0.5 (termed as half-minimum perturbation) for the terminal as well as hybrid states (*Figure 6C*). As expected, the half-minimum value for the terminal states is always higher than the same for the hybrid states (all five EMP networks lie below the x = y diagonal in *Figure 6C*). Interestingly, the 57N113E network that had a lower team strength among the EMP networks showed slightly higher resilience for hybrid states as compared to the other EMP networks, suggesting that as the strength of teams decreases, the difference between hybrid and terminal states in their stability tends to reduce. This argument is further strengthened by analysis of random networks (with Ts often less than that in WT networks), showing a comparatively higher resilience of hybrid states (as visualized by the distance from the x = y diagonal) (*Figure 6—figure supplement 1C*). Put together, these results endorse that 'teams' of nodes can also play a crucial role in maintaining a terminal phenotype, even when multiple nodes within a network are disrupted. Given the clear difference in the dynamic stability of hybrid and terminal phenotypes, we wanted to understand the biological implications of the dynamic stability characteristics and the presence of 'teams.

The multinode perturbation experiment in *Figure 6B* can be perceived as a state transition-inducing signal. The higher the extent of perturbation, the stronger the signal. Given this interpretation, we interrogated whether the transition trajectory/characteristics depend on the presence of teams in the network. To better quantify these dynamics, we performed an experiment mimicking a population of 100 cells exposed to a transitory signal. Each cell in the population starts from one steady state, and a random perturbation of a certain extent (varying from no node from perturbed [0] to all nodes being perturbed together [1]) is given to each cell. The state of the cell is then allowed to evolve until a steady state is reached. We repeat this experiment 10 times for each steady state of a network and perform statistical analysis. To measure the phenotypic transition, we made use of the Hamming distance between the initial (unperturbed) steady state and the steady state obtained after perturbation. The Hamming distance between two states is calculated as the fraction of nodes having different expression levels between the two states. The Hamming distance varies between 0 and 1, where a Hamming distance of 0 indicates identical states and a Hamming distance of 1 indicates states with all nodes having opposite levels of expression. Therefore, the states belonging to the same biological phenotype will be separated by a relatively smaller Hamming distance (close to 0) due to the presence of 'teams'.

The terminal phenotypes of EMP networks show a sigmoidal transition curve in terms of the mean Hamming distance across all replicates (*Figure 6Di*). In other words, at low levels of perturbation, the Hamming distance and change in EMT score remains low, suggesting that 'teams' offer 'resistance' to the signal, leading to a minimal change in the corresponding phenotype. Similarly, at high levels of perturbation, we see a complete change in the phenotype, as measured by the Hamming distance (being close to 1) and a drastic change in the EMT score. At intermediate levels of perturbation, a near-linear transition of phenotype is seen. We further quantified the phenotypic distribution (% of states corresponding to E, H, and M phenotypes) in these three levels of perturbation (as demarcated in *Figure 6Di*). As expected, for a population of cells starting from the epithelial phenotype, at low levels of perturbation, a majority of the cells remain epithelial, with a very small fraction converting to mesenchymal. At moderate levels of perturbation, we see an equal fraction of cells in epithelial and mesenchymal phenotypes, with a minor fraction switching to hybrid. At high levels, almost all cells turn mesenchymal. To identify how unique this sigmoidal pattern is to an EMP network (containing 'teams' of nodes), we evaluated this transition trajectory for a hypothetical/random network with low team strength (*Figure 6—figure supplement 1D*). We classify the steady states of random networks as terminal if all the active core nodes in the state belong exclusively to one of the two teams observed in that network. A team in a random network is identified as epithelial if the number of microRNAs

in the team is higher than that in the other team of the network. Unlike the case with EMP networks, the distinction between terminal and hybrid phenotypes in terms of their transition characteristics mostly disappears, and all the phenotypes have near-linear transition characteristics when perturbed (*Figure 6Ei*, *Figure 6—figure supplement 1Ei,ii*). Consistently, the corresponding phenotypic distributions at low, medium, and high levels of perturbations look comparable, irrespective of the initial phenotype (*Figure 6Eii*, *Figure 6—figure supplement 1Eiii*). This difference between the behavior of WT networks and random networks indicates that the teams govern transition trajectories emanating from various terminal or hybrid phenotypes in a network.

To better understand the dependence of the transition dynamics on the team strength, we quantified the area under the curve (AUC) for phenotypic distributions for various levels of perturbations

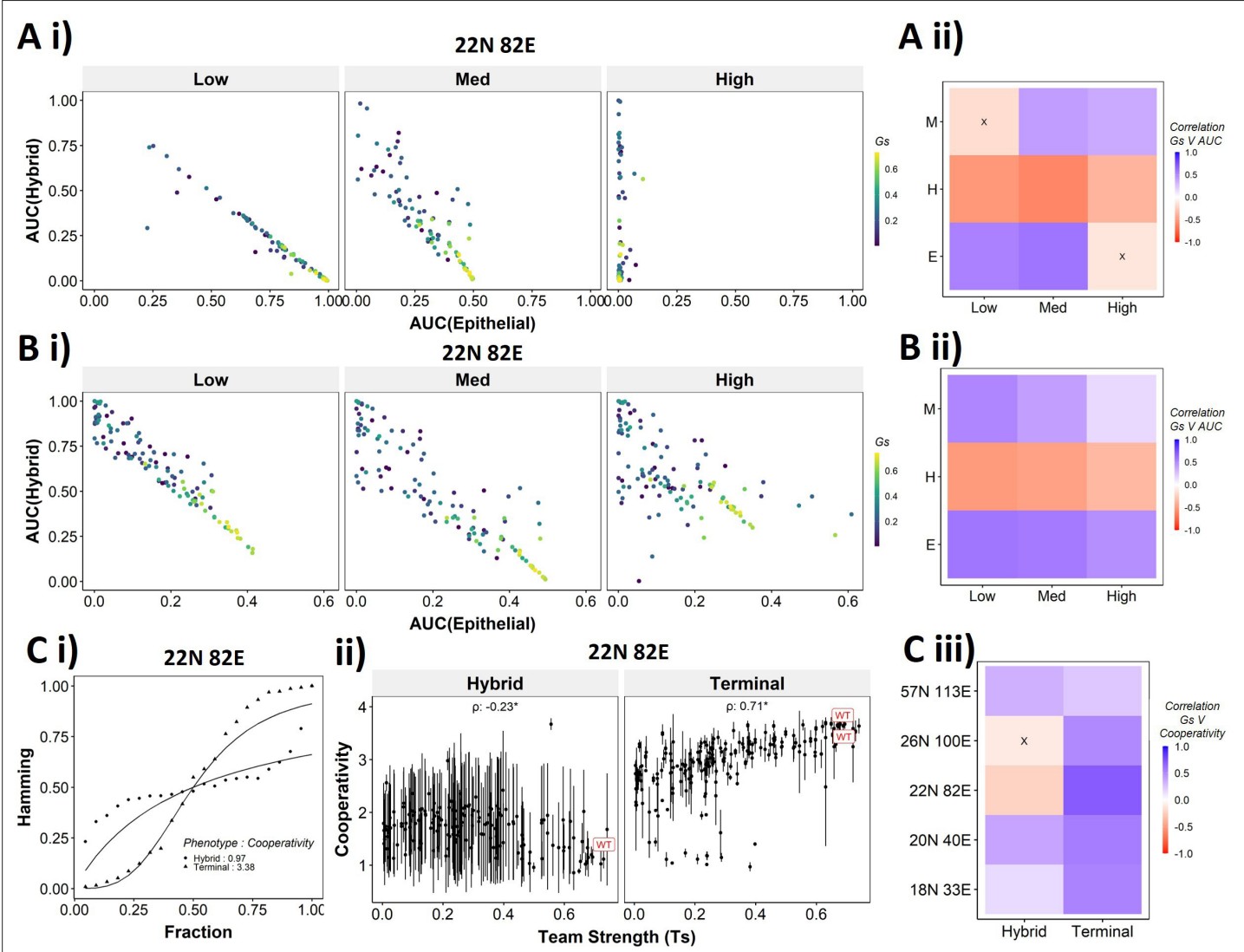

**Figure 7.** Distinction between the transition dynamics of hybrid and terminal phenotypes is lost as teams weaken. (**A**) (i) Scatterplot between mean area under the curve (AUC) of epithelial states and mean AUC of hybrid states when starting from epithelial phenotype. Each dot is a random network, colored by its team strength. (ii) Heatmap depicting the Spearman's correlation between team strength and AUC for the final phenotype (y-axis) and the regions in the sigmoidal plot (x-axis). (**B**) Same as (**A**), but for hybrid as the starting phenotypes. (**C**) (i) Depiction of cooperativity with a terminal and hybrid state of the wild-type (WT) epithelial–mesenchymal plasticity (EMP) network 22N 82E. (ii) Team strength vs. cooperativity for random and WT networks of size 22N 82E. Each dot represents the mean cooperativity for one network. The bars show standard deviation. (iii) Correlation between team strength and cooperativity for random networks corresponding to WT EMP networks of different sizes.

The online version of this article includes the following figure supplement(s) for figure 7:

**Figure supplement 1.** Dependence of perturbation coherence on team strength.

(for panels shown in *Figure 6Dii,Eii*). When starting from an epithelial phenotype, the networks with high team strength show low frequencies of the hybrid states as compared to the networks with low team strength, irrespective of the degree of perturbation made (*Figure 7Ai*), suggesting that the presence of 'teams' does not enhance the frequency of hybrid states. A quantification across the epithelial states for random networks as well revealed a negative correlation of the hybrid AUC with team strength at all levels of perturbation (*Figure 7Aii*). The correlation is positive with mesenchymal AUC at high perturbation and positive with epithelial AUC at high perturbation as expected since these are the dominant final (terminal) states in each case, respectively. Similar trends were seen for cells starting from mesenchymal phenotypes (*Figure 7—figure supplement 1A*).

The AUC analysis for cases with hybrid states as the initial conditions revealed similarly consistent trends. At low and high levels of perturbation, as the team strength of the network increases, the AUC of epithelial approaches 0.4 and that of hybrid approaches 0.1. At medium levels of perturbation, epithelial AUC nears 0.5, and hybrid AUC nears 0 for the maximum team strength network (*Figure 7Bi*). Unlike the trajectories seen for terminal phenotypes, because there is no preferred phenotype here in terms of either epithelial or mesenchymal, team strength correlated positively with both epithelial and mesenchymal AUCs and negatively with hybrid AUC (*Figure 7Bii*). These trends between the AUC and team strength were consistently seen across networks of all sizes (*Figure 7—figure supplement 1B*). Together, these observations indicate that the presence of two teams – one composed of epithelial master regulators and the other composed of mesenchymal master regulators – may reduce the frequency and stability of hybrid E/M phenotypes.

While the AUC analysis provided a good understanding of the transition properties, the mean Hamming distance plots intuitively demonstrated the difference in dynamic characteristics between the terminal and hybrid phenotypes of WT networks and that of random networks well. Hence, we quantified the sigmoidal nature of these transition curves by fitting them to a simple Hill's function and calculating the coefficient of cooperativity (n) for each such fit (*Figure 7Ci*). We then compared the

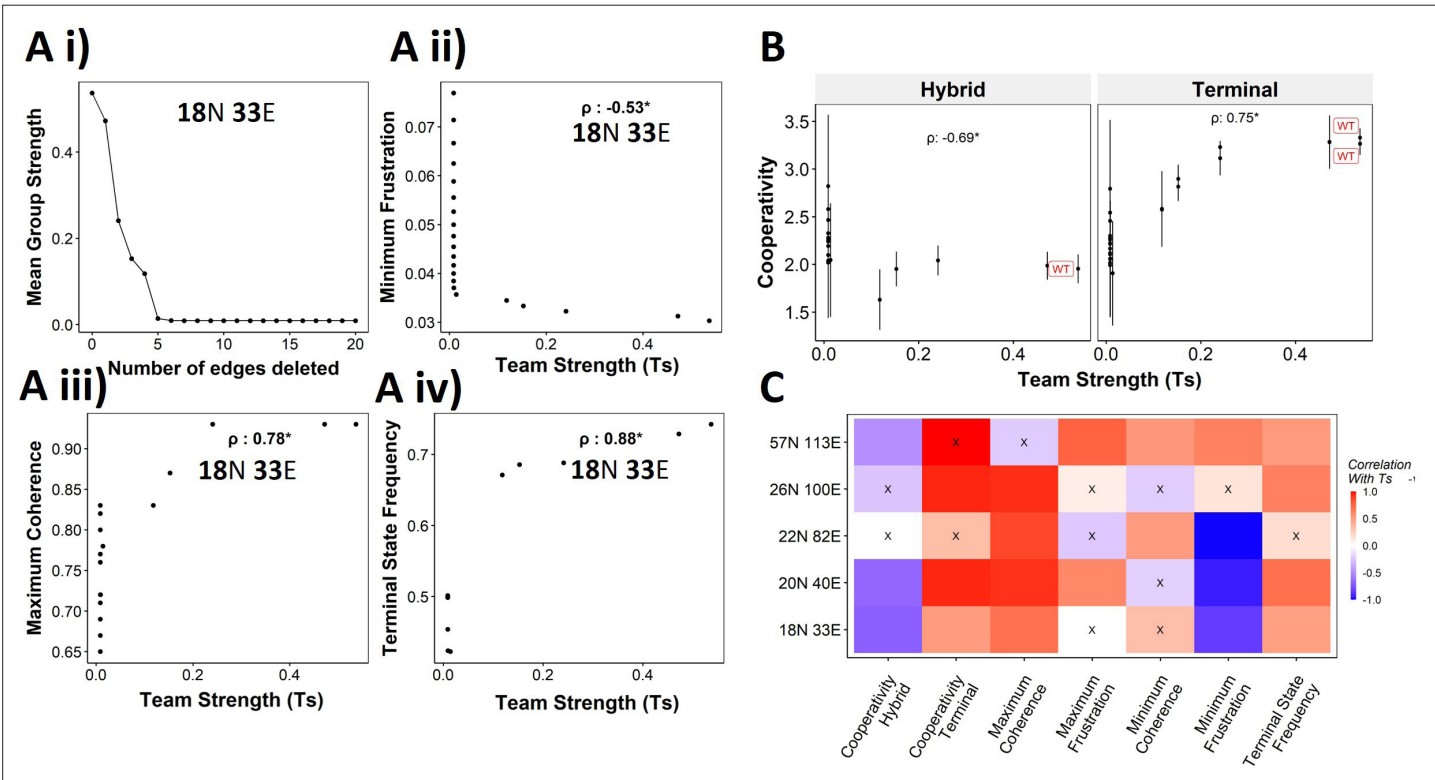

**Figure 8.** Reducing team strength leads to reduction in terminal phenotype stability. (**A**) (**i**) Lineplot to demonstrate the reduction of team strength with each edge perturbed. Change in (**ii**) minimum frustration, (**iii**) maximum coherence, and (**iv**) terminal state frequency with increase in team strength. (**B**) Mean cooperativity for the perturbed networks against the team strength. (**C**) Correlation between team strength and stability metrics for the perturbed networks (edge deleted one at a time sequentially as shown for 18N 33E network in panel **A**) obtained from all five epithelial–mesenchymal plasticity (EMP) networks.

mean cooperativity coefficient for trajectories obtained for terminal phenotypes and hybrid phenotypes of all random and WT networks against the team strength of the networks. The cooperativity coefficient of terminal phenotypes increased with increasing team strength (*Figure 7Cii*). The correlation of the mean cooperativity coefficient corresponding to terminal phenotypes with that of team strength was consistently positive across networks of different sizes, while no such trend was observed for hybrid phenotypes (*Figure 7Ciii*). The higher the value of n, the more step-like the corresponding sigmoidal function is. Thus these results endorse that the presence of teams makes the transition dynamics highly nonlinear and confers initial 'resistance' to exit a terminal phenotype (lag phase of sigmoidal curves).

Therefore, using the random networks as case studies, we were able to establish that the 'teams'' structure supports the terminal phenotypes dynamically and leads to unique dynamic transition signatures of the terminal and hybrid phenotypes in these different EMP networks.

## Targeted reduction of team strength in a network can reduce stability and robustness of EMP phenotypes

We have established the connection between the 'teams" structure and stability of terminal phenotypes in a correlative manner across multiple EMP networks of varying sizes. To propose a causative connection, we performed in silico edge deletion experiments in the WT EMP networks. In a given network, we ranked different edges in terms of their deletion, being able to maximize a reduction in team strength after deleting that edge. Sequential deletion of edges in this manner was performed to bring down the team strength to lower values (*Figure 8Ai*). The networks thus obtained saw a reduction in the stability of the terminal phenotypes, as measured by minimum frustration (*Figure 8Aii*), maximum coherence (*Figure 8Aiii*), and the frequency of terminal phenotypes (*Figure 8Aiv*). The dependence of the stability of terminal phenotypes on team strength is especially prominent in the areas corresponding to the initial linear regime (number of edges deleted <5 in *Figure 8Ai*), where the team strength falls sharply. The estimated cooperativity coefficient (n in Hills function) of the transition dynamics of the terminal and hybrid phenotypes also showed expected trends: the higher the team strength, the higher the cooperativity (i.e., the more sigmoidal the curve) (*Figure 8B*). A summary of the correlations obtained between Ts and various metrics is given as a heatmap in *Figure 8C*. We see a positive correlation between Ts and measures of stability of terminal phenotypes (cooperativity, terminal state frequency). Maximum coherence and minimum frustration showed consistent trends (positive and negative correlation with Ts, respectively) in four and three out of five networks, respectively. While the inconsistency for the 57N 113E network could be due to the low team strength of the network (*Figure 4—figure supplement 2B*), that in the 26N 100E network could possibly be due to the high frustration in the network (*Figure 3—figure supplement 1D*). These results suggest that as long as the structure of 'teams' is maintained, terminal phenotypes remain dominantly stable, and their stability can be predicted by the strength of 'teams,' which can be calculated from the influence matrix alone, without performing any dynamic simulations.

## Teams stabilize terminal phenotypes in SCLC and melanoma networks as well

We next asked whether teams can be seen in other cell-fate decision networks. As mentioned earlier, we had seen such teams in the SCLC network (*Chauhan et al., 2021*). Similarly, teams of nodes have been seen in a regulatory network underlying melanoma that has been shown to be capable of driving phenotypic heterogeneity in melanoma (*Figure 9A*, *Pillai and Jolly, 2021*). We wanted to check whether the effect of teams on the stability of phenotypes is extendable beyond EMP. First, we looked at the phenotypes emergent from SCLC and melanoma networks and found that in SCLC there are two classes of states, the terminal states with high stability and hybrid states with low stability. Interestingly, the melanoma network only resulted in terminal states (*Figure 9—figure supplement 1A and B*). Similar to EMP networks, we generated random networks from the SCLC and melanoma networks. The team strength of the WT networks, while a lower value as compared to that of WT EMP networks, was higher than most of the corresponding random networks (*Figure 9B*). Furthermore, high team strength networks showed high maximum stability of the networks (*Figure 9C*). Furthermore, we studied an EMP network obtained by combining the five EMP networks analyzed here and another related EMP network (8N 17E, *Hong et al., 2015*). We were able to clearly identify teams in

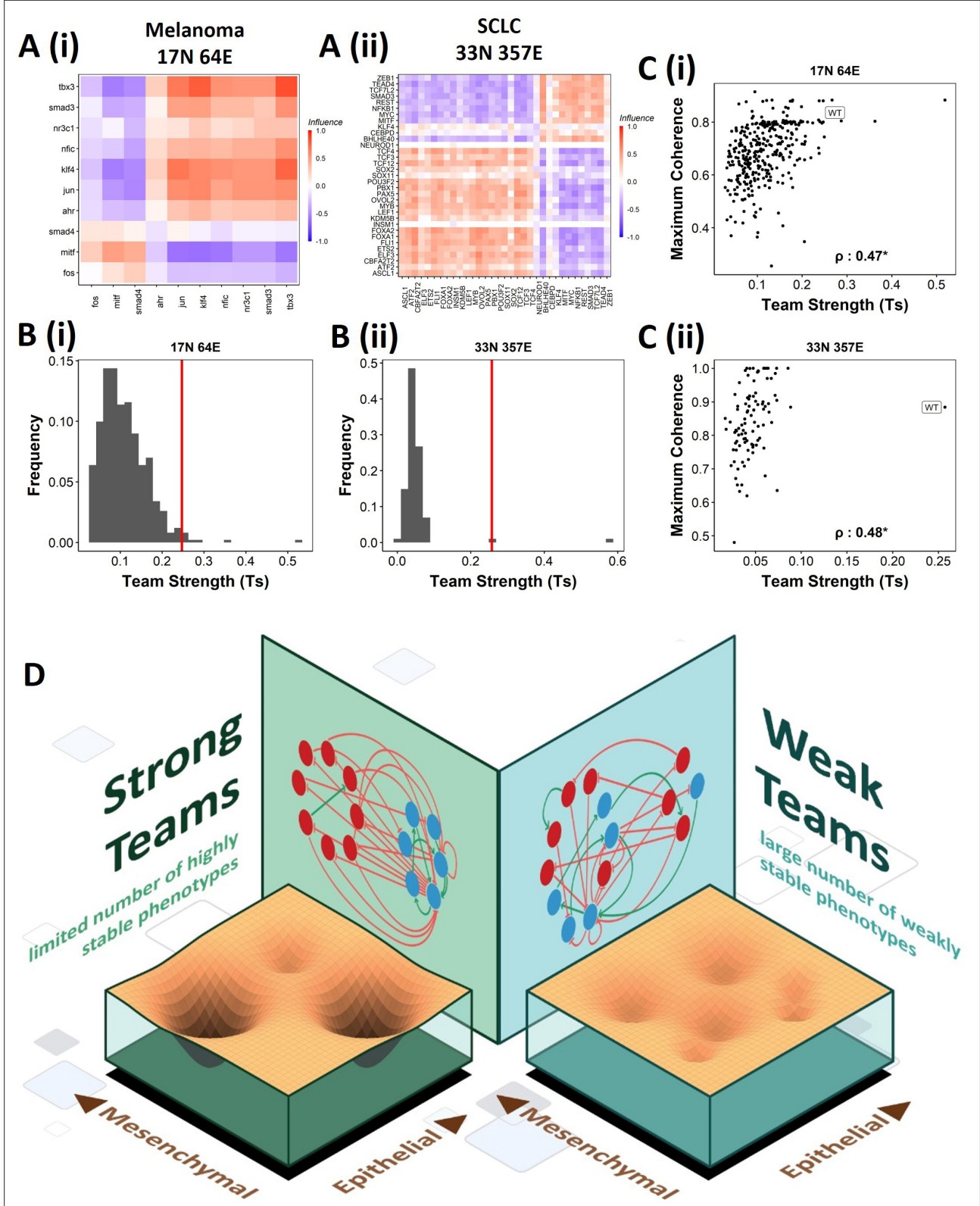

**Figure 9.** Effect of teams on phenotypic landscape in small cell lung cancer (SCLC) and melanoma. (**A**) Influence matrices of (**i**) melanoma (17N 64E) and (**ii**) SCLC (33N 357E) networks, depicting the two team structure observed. (**B**) Comparison of team strength distributions obtained for random networks corresponding to (**i**) melanoma and (**ii**) SCLC, with the wild-type (WT) melanoma and SCLC team strengths labeled by red vertical line. (**C**) Scatterplots depicting the maximum coherence against the team strength of the random networks corresponding to (**i**) melanoma and (**ii**) SCLC. Spearman's

*Figure 9 continued on next page*

Figure 9 continued

correlation coefficient reported. *p<0.05. (**D**) Schematic showing the effect of team structure on the phenotypic stability landscape emergent from the network topology.

The online version of this article includes the following figure supplement(s) for figure 9:

**Figure supplement 1.** SSF and correlation analysis of non-EMP netowrks.

this combined network (*Figure 9—figure supplement 1C*), and the influence matrix was very similar to the correlation matrix (*Figure 9—figure supplement 1D*). Again, the terminal phenotypes (characterized based on the team structure) had high SSF (*Figure 9—figure supplement 1E*). Together, these observations imply that the role of teams of nodes in determining the stability landscape can be an extendable phenomenon to contexts other than EMP. Specifically, strong teams improve the stability of terminal phenotypes, leading to a strongly bimodal phenotypic stability landscape (*Figure 9D*).

## Discussion

Cellular decision making often involves a limited number of specific fates the cells can switch among. However, the underlying regulatory systems appear disproportionately more complex in many cases. In this study, we show how having teams of nodes is one way the networks achieve this property of having a limited number of phenotypes despite their size and complexity.

One of our primary goals was to identify 'teams' of nodes in the network using unsupervised mechanisms, which we could accomplish by using hierarchical clustering on the influence matrix. We found that the team strength (Ts) was higher for the WT EMP networks than the corresponding random networks generated by shuffling the edges between the nodes. One way to interpret this trend is that the random networks are evolutionary alternatives that could have happened using the same nodes and edges. However, the biological networks were selected to optimize for the strength of 'teams.' Similar analysis has been employed while understanding other properties in previous studies (*Hari et al., 2020*; *Tripathi et al., 2020b*; *Hebbar et al., 2022*). The fact that the WT network topologies do not have the absolute highest values for these properties is a recurring theme: the reasons currently remain unclear (*Hebbar et al., 2022*; *Hari et al., 2020*).

We defined the stability of the steady states emergent from these networks in two ways: SSF as a measure of global stability and coherence as a measure of local stability. Additionally, we used frustration as a measure of the agreement of the network topology with a given state. Previous studies have shown that the epithelial and mesenchymal phenotypes show a higher SSF and lower frustration. In contrast, hybrid phenotypes show the opposite trend, and our results echo these findings (*Font-Clos et al., 2018*; *Tripathi et al., 2020a*). An important observation, however, was that this strong antagonistic relationship between SSF and frustration is only maintained in the presence of strong teams.

The 'teams' we found are reminiscent of modularity in large-scale networks identified in previous studies. Particularly, two definitions of modularity are relevant. One is the existence of communities of nodes in the network that each performs a unique set of functions and has limited interactions with the other communities in the network (*Wang and Albert, 2013*). The two teams found in EMP networks do perform unique functions by supporting two mutually exclusive phenotypes (epithelial and mesenchymal). However, extensive interactions between these teams are also present, enabling a strong toggle switch between the teams. Another instance of modularity is the presence of strongly connected components (*Zañudo and Albert, 2013*; *Steinway et al., 2015*). This structure enables an efficient transfer of information from the signal given to any of the nodes to all nodes, thereby showing coordinated expression of all components. While the teams do have the property of coordinated expression, connectedness is not a necessary condition. The nodes of the epithelial team in the 22N 82E network in our study do not have any interaction within themselves but have coordinated expression only through the mutually inhibiting loops they form with the nodes of the mesenchymal teams. This apparent discrepancy can be attributed to the epithelial nodes constituting mostly microRNAs and the exclusion of recently reported molecular TFs such as KLF4 and ELF3 (*Subbalakshmi et al., 2022*; *Subbalakshmi et al., 2021*).

One significant finding of our study is identifying terminal and hybrid phenotypes purely based on the network topology, that is, without having to simulate the network dynamics. While the biochemical composition of epithelial and mesenchymal phenotypes is relatively well defined across different cell

types, the definition of hybrid phenotypes is quite context-dependent (*Jolly et al., 2016*; *Steinway et al., 2015*; *Watanabe et al., 2019*). One reason for this difference could be the nonuniformity in the gene lists used to identify these phenotypes from high-throughput data (*Puram et al., 2017*; *Pastushenko et al., 2018*). While our method of identification of hybrid phenotypes only considers the limited number of nodes available in the networks, the method itself depends on how these nodes interact and not the nodes themselves. Hence, if we can infer the regulatory network from a set of genes in a cell type, using the teams' approach, we can better define hybrid phenotypic signatures.

While we specifically do not present any experimental evidence for teams in EMP, a recent analysis based on pairwise correlations between master regulators of EMT/MET has revealed that MET inducers positively correlate with one another but negatively with EMT inducers (*Chakraborty et al., 2021*; *Jia et al., 2020*). Teams have also been seen in melanoma (*Pillai and Jolly, 2021*) and SCLC (*Chauhan et al., 2021*) where network topology and correlation matrices obtained from experimental data are quite consistent in decoding how nodes act together to decide certain phenotypes. Indeed, we find that teams contribute to the phenotypic landscape similarly to EMP networks. Thus, our results based on network topology can offer a possible mechanistic reason why such teams are seen in correlation matrices for both in vitro and in vivo data. Recent studies have started to identify the connections between different axes of plasticity underlying cancerous cells, such as EMP, drug resistance, immune evasion, and dormancy. While, in most cases, these connections have been explored using small-scale networks, the presence of teams provides an intuitive way of reducing the complexity of large networks and therefore enables the coordination of multiple axes of plasticity at a large scale; for example, EMP and drug resistance (*Sahoo et al., 2021*).

Our analysis suggests that the hybrid phenotypes are 'metastable,' similar to previous reports (*Font-Clos et al., 2018*). Experimentally speaking, the frequency and stability of hybrid E/M phenotypes seem to be quite varied (*Pastushenko et al., 2018*; *Ruscetti et al., 2016*; *Jolly et al., 2016*). This observation leads to a hypothesis whether a third team comprising factors driving a partial phenotype is required to stabilize hybrid E/M phenotypes, such as NP63 (*Dang et al., 2015*) and NRF2 (*Bocci et al., 2019*). Three teams of players have been proposed to give rise to three distinct phenotypes, for instance, in the case of CD4+ T cell differentiation, where the three master regulators (and their corresponding team members) inhibit each other, driving Th1, Th2, and Th17 phenotypes (*Zhu et al., 2010*; *Duddu et al., 2020*). Another reason underlying the higher stability of hybrid E/M phenotypes may be cell–cell communication (*Jolly et al., 2015*) and/or epigenetic regulations (*Jia et al., 2021*), both of which have not been included in our analysis.

Teams of nodes forming a toggle switch can be an excellent way to explain the canalization property observed in development. The presence of teams can make the cell-fate decision robust to multiple environmental fluctuations and biochemical cues (*Figure 6*). However, should teams be seen only in the most differentiated cell states in a lineage, or can they stabilize intermediate cell fates too? If so, are teams retained upon further differentiation or disrupted at a structural level? How would the teams at different levels of differentiation interact with each other? Furthermore, how do teams bifurcate upon sequential instances of cellular differentiation (*Zhou and Huang, 2011*)? If two teams of nodes determine the decision between phenotype A and phenotype B. When B further differentiates to phenotypes C and D, do new teams supporting C and D emerge, or does team B break down into two sub-teams? Identifying changes in regulatory networks that would be required to implement these rearrangements will be an exciting future direction.

Here, we see teams in terms of intracellular regulatory networks. However, this framework of identifying the composition of two (or more) teams acting together to reinforce each other in scenarios of competing outcomes can be applied more broadly, particularly in multicellularity. The emergence of multicellularity (e.g., development of tissues and organs) has been proposed via the establishment of biochemical cooperation between individual cells (*Kaneko, 2016*). Such multicellularity and cooperation are often observed in cancer. As tumors grow in size, the cells in different spatial locations start specializing in different functionalities such as metabolism or cell division to compensate for hurdles such as hypoxia, leading to the survival of the tumor as a whole. Different cells in a microenvironment can also form teams with pro-tumor and anti-tumor influence (*Capp et al., 2021*). Similarly, cancer cells undergoing metastasis form clusters that lead to better survival than that of individual CTCs, owing to the heterogeneity in the clustered CTCs equipped to deal with different hurdles, providing the cluster a better chance of survival (*Aceto et al., 2014*; *Hong et al., 2016*). All of these scenarios

can be viewed as multiple teams/groups of cells interacting with each other, each specializing in one or the other functionality essential for the survival of the population as a whole. Extending our ideas of team strength, we would expect populations with stronger interactions to have better chances of survival. However, extensive analysis is needed to make such claims as there are apparent differences in the formalisms that must be addressed. First, in most cases of multicellularity, the participating cells exhibit plasticity, such that they can change their phenotype/identity dynamically depending on the environment (*Xiao et al., 2022*; *Bhattacharya et al., 2021*). Such plasticity has been theorized to be instrumental in the emergence and survival of multicellular beings (*Alvarez et al., 2022*). In our analysis, the nature of the nodes has been conserved, and hence the impact of such dynamism on network structure is unclear. Second, the number of cells keeps changing, adding another dynamic aspect to the network structure. Third, it is not clear what kind of interactions happen within a team of cells, which is a part of the team strength inferred in GRNs. Furthermore, the interaction across teams also need not always be cooperation. In the emergence of therapy resistance, studies have shown in certain contexts in the absence of therapy that resistant cells support the growth of sensitive cells while sensitive cells inhibit the growth of resistant cells (*Nam et al., 2021*). The dynamic network issues can be addressed by considering a network with nodes as homogeneous subpopulations (i.e., teams) within a population rather than individual cells. The level of activity of a node will then be the number of cells in the subpopulation. This interpretation does assume that the team acts as a single component, which has to be validated for different biochemical and spatial interactions of cells within a team.

Overall, our study highlights that despite their apparent complexity, design principles are hidden in the topology of cell-fate decision-making biological networks that can canalize phenotypic repertoire and shape the corresponding emergent phenotypic landscape (*Figure 9D*). Insights gained from such network topology and/or geometric approaches (*Sáez et al., 2022*; *Rand et al., 2021*) to studying gene regulatory networks can contribute to accurately identifying the underlying landscape and modulating it for cellular reprogramming purposes.

## Methods
### Notations
The following notations are followed throughout the article unless mentioned otherwise:

- $N$ : number of nodes in a network
- $E$ : number of edges in a network
- *Adj* : adjecency matrix; also referred to as the interaction matrix
- *Infl* : influence matrix
- $i, j$ : indices that refer to the nodes in the corresponding positions in adjacency or influence matrices
- $Adj_{ij}$ : the interaction strength of the edge from $i$th node to the $j$th node

$$\text{Adj}_{ij} = \begin{cases} +1, i -> j \\ -1, i - \dashv j \\ 0, otherwise \end{cases} \tag{1}$$

Similarly, $Infl_{ij}$ is the influence of the $i$th node on the $j$th node
- $S(t)$ : state of a network at time $t$
- $s_i(t) \in \{-1, 1\}$ state/activity of a node of a network at time $t$

$$S(t) = \{s_i(t)\}; \ i \in 1, 2, ..., N \tag{2}$$

- *Sig* : the set of signal nodes.
- *Out* : the set of output nodes
- *Core* : the set of core nodes

## Figure 1

### Network visualization

Cytoscape 3.9.0 (*Shannon et al., 2003*) was used to visualize the networks studied. Edges were colored based on the sign (inhibiting and activating), and nodes were colored based on their nature (epithelial, mesenchymal, and peripheral).

### Boolean simulations

Boolean modeling is a logic-based, simple and discrete system for capturing the dynamics of biological networks. The framework describes each node of the network as a binary variable (–1 or 1) by considering a threshold value or quantity of the molecule that can elicit the necessary downstream function. In the framework used in this study (*Font-Clos et al., 2018*), a state of a network is defined by a binary string of variables $s_i$, which gives information about which node is active/ON ($s_i = 1$) or inactive/OFF ($s_i = -1$). The interactions between the nodes are represented using the nonsymmetric adjacency matrix $Adj$, where each element of the matrix, $Adj_{ij}$, is the interaction strength of the edge from $i$th to $j$th node of the network. All activations are given a weight of 1, and all inhibitions are given weight of –1. The simulations are conducted asynchronously (one randomly chosen node is updated at each iteration). The state of the system is updated using a majority rule given below:

$$s_i(t + 1) = \begin{cases} +1, \sum_j Adj_{ij}s_j(t) > 0 \\ -1, \sum_j Adj_{ij}s_j(t) < 0 \\ s_i(t), \sum_j Adj_{ij}s_j(t) = 0 \end{cases} \tag{3}$$

Simulations are carried out until either of the two conditions is reached: (1) t > 1000 or (2) $s_i(t + 1) = s_i(t) \, \forall \, i \in \{1, ..., N\}$. The latter condition implies that a steady state has been reached. $S(t)$ is identified as the steady state.

### Steady-state frequency (SSF)

To obtain SSF, we simulate the network with multiple randomly chosen initial states and count the fraction of such simulations that end in a given steady state.

### Random network generation

The generation of random networks is an important technique that enables us to analyze the similarities and/or differences between biological networks and networks that do not occur in nature (essentially 'random'). To create random networks, we start with a biological network, select a pair of edges randomly in the network, and swap the nature of the edges (see *Figure 1B*). This exercise was repeated k times to create random networks. We found that larger values of k lead to the random networks having very low team strength. Hence, to capture networks with moderate levels of team strengths as well, we chose k = 10. This scheme conserves node degree, activatory and inhibitory contacts, and the number of nodes activated and inhibited.

## Figure 2

### Coherence

Coherence is calculated by perturbing the node expression levels in a state. 'Perturbation' in the context of Boolean networks corresponds to essentially flipping or reversing a state. For instance, if the level of $S_i$ is ON (1), then perturbing it entails switching it to OFF (–1). The general form is given below:

$$s_i^{pert} = \begin{cases} 1, s_i = -1 \\ -1, s_i = 1 \end{cases} \tag{4}$$

where $n_i^{pert}$ is the perturbed node. Node perturbations were done for every node in every steady state one at a time. For a steady state S, coherence is defined as the probability that simulation

followed by a single node perturbation of that state would result in the original steady state. To calculate the coherence of a steady state, we perturb the state at one node at a time to simulate the network with the perturbed state as the initial condition. For each simulation, we assign a score of 1 if the original state is achieved, and 0 if it is not. We repeat these simulations for each node in the network for K = 100 iterations and define coherence as the average of the assigned scores over all simulations as follows:

$$Coherence_S = \frac{\sum_{k=1}^{100} \sum_{i=1}^{N} \begin{cases} 1, S^{pert} = S \\ 0, S^{pert} \neq S \end{cases}}{N * K} \tag{5}$$

where $S_i^{pert}$ is the steady state obtained after simulation of the perturbed steady state, and N is the number of nodes in the network.

## Bimodality coefficient

Bimodality coefficients have been calculated using the following formula (**Knapp, 2007**):

$$BC = \frac{m_3^2 + 1}{m_4 + 3\frac{(n-1)^2}{(n-2)(n-3)}} \tag{6}$$

where $n$ is the number of observations, $m_3$ is the skewness, and $m_4$ is the kurtosis of the distribution of the metric of interest.

# Figure 3

## Frustration

Frustration is a measure of the agreement between the network topology and a given steady state. For a given network and state, the frustration is calculated as follows:

$$\frac{\sum_{i,j=1}^{N} \begin{cases} 1, Adj_{ij}s_i s_j < 0 \\ 0, otherwise \end{cases}}{E} \tag{7}$$

where $E$ is the number of edges in the network.

# Figure 4

## Influence matrix

The influence matrix, as the name suggests, is a matrix where each element at ($i,j$) position records the influence of $i$th node on the $j$th node in the network. This influence is mediated through one or more serially connected edges that form a path from the $i$th node to the $j$th node in the network. Path length ($l$) is defined as the length of such paths being considered for the calculation of influence. As a result, for a path length of 1, the influence matrix is equivalent to the adjacency matrix $Adj$. For a path length of $l$, the influence is calculated as $Adj^l$. Similarly, the influence is calculated for all path lengths up to a maximum path length of $l_{max} = 10$ edges. Finally, the influence matrix for a path length of $l_{max}$ is calculated by the following equation (**Chauhan et al., 2021**):

$$Infl_{max} = \frac{\sum_{l=1}^{l_{max}} \frac{Adj^l}{Adj^l_{l_{max}}}}{l_{max}} \tag{8}$$

where $Adj_{max}$ is derived by setting all nonzero entries of the adjacency matrix to 1, and is thus utilized as the normalizing factor. The division $\frac{Adj^l}{Adj^l_{max}}$ is element-wise:

$$\frac{Adj^l}{Adj^l_{max}}(i,j) = \begin{cases} 0, \ Adj^l_{max}(i,j) = 0 \\ \frac{Adj^l(i,j)}{Adj^l_{max}(i,j)}, \ otherwise \end{cases} \tag{9}$$

The division with $l_{max}$ ensures that the elements of $Infl_{l_{max}}$ are constrained between –1 and 1.

## Identifying teams and calculating team strength

In a given network, a set of 'core' nodes $T$ are said to be a team if

$$Infl_{ij} > 0 \forall i, j \in T$$

Additionally, we find that nodes belonging to different teams influence each other negatively. The algorithm used to identify such teams in the influence matrix is as follows:

Once the two teams are obtained, the team strength of a network is calculated as follows:

$$T_{kl} = \frac{\sum_{i \in T_k, j \in T_l} Infl_{ij}}{n_{kl}}, \ k, l \in \{1, 2\} \tag{10}$$

$$T_S = \frac{\sum_{k,l \in \{1,2\}} |T_{kl}|}{4} \tag{11}$$

where $T_1$ and $T_2$ are the two teams of nodes identified using hierarchical clustering, and $n_k l$ is the product of the number of nodes in $T_k$ and $T_l$. To classify the teams as epithelial or mesenchymal, we counted the number of microRNAs present in each team. Because the microRNAs in the five EMP networks considered here are exclusively epithelial, we labeled the team that has the highest number of microRNAs as the epithelial team.

## Distance between influence and interaction matrices

The distance between influence and correlation matrices was calculated using the following formula:

$$d = \sum_{i=1}^{N} \sum_{j=1}^{N} \frac{|Cor_{ij} - Infl_{ij}|}{2N^2} \tag{12}$$

where $Cor$ is the correlation matrix obtained over Boolean simulations.

## State strength

The strength of a state S is similar to frustration in that it calculates the support of the influence matrix to the state and is calculated using the following formula:

$$Strength_S = \sum_{i,j=1}^{N} Infl_{ij} s_i s_j, \ s_i, s_j \in \{0, 1\} \tag{13}$$

## Figure 6

## Single-node perturbation

For single-node coherence (*Figure 6A*), we perturb the steady states one node at a time, resulting in the following calculation:

$$Coherence_{S_i} = \frac{\sum_{k=1}^{100} \begin{cases} 1, S^{pert} = S \\ 0, S^{pert} \neq S \end{cases}}{K} \tag{14}$$

where $Coherence_{S_i}$ is the coherence of steady state $S$ when node $i$ is perturbed. We further calculate the coherence specific to a set of core nodes upon perturbing signal nodes (*Figure 6—figure supplement 2*).

$$\text{Coherence}_{S_i} = \frac{\sum_{k=1}^{100} \begin{cases} 1, S_{Core}^{pert} = S_{Core} \\ 0, S_{Core}^{pert} \neq S_{Core} \end{cases}}{K}, i \in Sig \tag{15}$$

where, $S_{Core}$ is the configuration of core nodes in the state $S$.

## Multinode perturbation

We performed experiments to characterize the stability of terminal and hybrid steady states over 'n' node perturbations, 'n' ranging from 1 to N (total number of nodes in a given network). Essentially, this means perturbing 'n' nodes at a time for each steady state, instead of perturbing just one node as seen for coherence calculations. For simplicity, the number of such perturbations for a particular value of 'n' was decided by the following rule:

$$Number\ of\ perturbations = min(100, \binom{N}{n}) \tag{16}$$

The perturbed steady states were then simulated using Boolean formalism for 1000 time steps, with 10 repeats to accommodate the fact that asynchronous Boolean simulations can allow a single initial condition to converge to multiple steady states. The final state obtained is compared with the original steady state by employing coherence and Hamming distance measures. The latter entails comparing these two states by considering the number of bit positions in which the two bits are different. The EMT score is also calculated for the final state obtained. We repeat these simulations for each 'n' number of perturbations in the network for K = 100 iterations and take the average of the final coherence, Hamming, and EMT score values for each original steady state.

## Statistical tests

### Percentile calculation

We calculate the percentile of the WT networks in the corresponding random network distribution for many metrics in this study. For a list of numbers $v$ that holds the measures of a given metric for random networks, and $W$ being the corresponding measure for the WT network, we calculate the fraction of members of $v$ less than $w$ and multiply the fraction with 100 to get the percentile.

### Correlations

All correlation analyses were done using Spearman's correlation method using '*cor.test*' function in *R 4.1.2*. The corresponding statistical significance values are represented by '*'s, to be translated as *$p < 0.05$.

### ANOVA

## Data and code availability

The codes used for generating the data (random networks and simulation), analyzing the data, and generating figures are made available as an R package at https://github.com/askhari139/Teams (*Hari et al., 2022* copy archived at swh:1:rev:fdf4f636f6762e6d2193d1bc71944d20a087bf3a). A detailed description of the codes has been included for ease of reproducibility.

## Acknowledgements

This work was supported by Ramanujan Fellowship awarded to MKJ by Science and Engineering Research Board (SERB), Department of Science and Technology (DST), Government of India (SB/S2/RJN-049/2018), by Infosys Young Investigator award to MKJ supported by Infosys Foundation, Bangalore, and by the Prime Minister's Research Fellowship awarded to KH. Atchuta Srinivas Duddu is acknowledged for artwork (*Figure 2B*, *Figure 9D*)

## Additional information

### Funding

| Funder | Grant reference number | Author |
|---|---|---|
| Science and Engineering Research Board | SB/S2/RJN-049/2018 | Mohit Kumar Jolly |

The funders had no role in study design, data collection and interpretation, or the decision to submit the work for publication.

### Author contributions

Kishore Hari, Data curation, Formal analysis, Methodology, Writing - original draft; Varun Ullanat, Archana Balasubramanian, Aditi Gopalan, Data curation, Formal analysis; Mohit Kumar Jolly, Conceptualization, Supervision, Funding acquisition, Methodology, Writing - review and editing

### Author ORCIDs

Kishore Hari http://orcid.org/0000-0001-5655-9039
Mohit Kumar Jolly http://orcid.org/0000-0002-6631-2109

### Decision letter and Author response

Decision letter https://doi.org/10.7554/eLife.76535.sa1
Author response https://doi.org/10.7554/eLife.76535.sa2

## Additional files

### Supplementary files
• Transparent reporting form

### Data availability

The current manuscript is a computational study. All raw numerical data used to generate the graphs is available at Dryad.

The following dataset was generated:

| Author(s) | Year | Dataset title | Dataset URL | Database and Identifier |
|---|---|---|---|---|
| Jolly MJ | 2022 | Data from: Landscape of epithelial–mesenchymal plasticity as an emergent property of coordinated teams in regulatory networks | http://dx.doi.org/10.5061/dryad.ncjsxksz7 | Dryad Digital Repository, 10.5061/dryad.ncjsxksz7 |

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

## Appendix 1

### Methods

RAndom CIrcuit PErturbaiton (RACIPE)

RACIPE (*Huang et al., 2017*) is a tool that simulates transcriptional regulatory networks (TRNs) in a continuous manner. Given a TRN, it constructs a system of ordinary differential equations (ODEs) representing the network. For a given node $T$ and a set of input nodes $P_i$ and $N_j$ that activate and inhibit $T$, respectively, the corresponding differential equation is given as *Equation 17*.

$$\frac{dT}{dt} = G_T * \prod_i \frac{H^{S+}(P_i, P_{iT}^0, n_{P_i,T}, \lambda_{P_i,T})}{\lambda_{P_i,T}} * \prod_j H^{S-}(N_j, N_{iT}^0, n_{N_j,T}, \lambda_{N_j,T}) - k_T * T \tag{17}$$

Here, $T$, $P_i$, and $N_j$ represent the concentrations of the species. $G_T$ and $k_T$ denote the production and degradation rates, respectively. $P_{iT}^0$ is the threshold value of $P_i$ concentration at which the nonlinearity in the dynamics of $T$ due to $P_i$ is seen. $n$ is termed as Hill coefficient and represents the extent of nonlinearity in the regulation. $\lambda$ represents the fold change in the target node concentration upon overexpression of regulating node. Finally, the functions $H^{S+}$ and $H^{S-}$ are known as shifted Hill functions (Lu2013) and represent the regulation of the target node by the regulatory node (*Equation 18*).

$$H^{S+/-}(B, B_A^0, n_{B,A}, \lambda_{B,A}) = \frac{B_A^{0\,n_{B,A}}}{B_A^{0\,n_{B,A}} + B^{n_{B,A}}} + \lambda * \frac{B^{n_{B,A}}}{B_A^{0\,n_{B,A}} + B^{n_{B,A}}} \tag{18}$$

Note that, for high values of the regulatory node concentration, $H^{S+/-}$ approaches $\lambda$. For the model generated in this way, RACIPE randomly samples parameter sets from a predefined set of parameter ranges estimated from BioNumbers (*Milo et al., 2010*). The default ranges used by RACIPE (*Huang et al., 2017*) are listed in *Appendix 1—table 1*.

**Appendix 1—table 1.** Parameter ranges for RACIPE simulations.

| Parameters | Minimum | Maximum |
|---|---|---|
| Production rate (G) | 1 | 100 |
| Degradation rate (k) | 0.1 | 1 |
| Fold Change (Inhibition $\lambda$) | 0.01 | 1 |
| Fold Change (Activation $\lambda$) | 1 | 100 |
| Hill coefficient | 1 | 6 |
| Threshold | The ranges depend on inward | regulation - half functional rule |

### Discretization of RACIPE output and calculating the state frequency

For a given network with $i = [1, n]$ nodes, the steady-state expression levels of the nodes were normalized about the mean and standard deviation across all parameter sets.

$$Z_i = \frac{E_i - \overline{E_i}}{\sigma_i} \tag{19}$$

where, for the $i$th node, $E_i$ is the steady-state expression level $\overline{E_i}$ is the combined mean and $\sigma_i$ is the combined standard deviation. The z-scores are then classified based on whether they are negative or positive into 0 (low) and 1 (high) expression levels, respectively. Each steady state of the network is thus labeled with a string of 1s and 0s, discretizing the continuous steady-state levels. We then calculate the total frequency of each discrete state by counting the occurrence in all the parameter sets. For parameter sets with n steady states, the count of each steady state is taken as 1/n, invoking the assumption that all the states are equally stable.

### Quantitative convergence

To estimate the optimal sample size of parameter sets for RACIPE and that of initial conditions for Boolean models, all networks were simulated at different sample sizes in triplicates and the mean

and variance of the SSF distribution was calculated. 10,000 was estimated as the ideal sample sizes for both methods as it was the smallest sample size for which the variance in steady-state frequencies was minimum and the mean of the same was consistently similar to that of higher sample sizes.

## Relation between frustration and stability of a state

In the ising Boolean formalism, the update rules are defined as follows:

$$s_i(t+1) = \begin{cases} +1, \sum_j Adj_{ij}s_j(t) > 0 \\ -1, \sum_j Adj_{ij}s_j(t) < 0 \\ s_i(t), \sum_j Adj_{ij}s_j(t) = 0 \end{cases} \tag{20}$$

with standard definitions of all involved terms. The update rules indicate that, for the activity of a node to remain conserved in the next time step,

$$s_i(t+1) = \begin{cases} \sum_j Adj_{ij}s_j(t) >= 0, s_i(t) = +1 \\ \sum_j Adj_{ij}s_j(t) <= 0, s_i(t) = -1 \end{cases} \tag{21}$$

In other words,

$$s_i(t+1) = s_i(t) \, if \sum_j Adj_{ij}s_j(t)s_i(t) >= 0 \tag{22}$$

An edge $ij$ is said to be frustrated for a given state (not necessarily steady state), if

$$Adj_{ij}s_i(t)s_j(t) < 0 \tag{23}$$

Note the similarity of the expressions in *Equation 22* and *Equation 23*. Now, consider the following two extreme cases of states:

- All edges are frustrated, that is, *Frustration* = 1. In this case, since $Adj_{ij}s_i(t)s_j(t) < 0$ for all $i, j$, the condition in 22 is never satisfied. Hence, none of the nodes in the state can retain their activity, and therefore the state cannot be a steady state.
- No edge is frustrated, that is, *Frustration* = 0. In this case, the condition in 22 is satisfied for all $i, j$ and hence for all nodes, making the state steady.

These extreme cases hence seem to indicate that higher the frustration, lesser the chance of a state being steady. However, having the frustration value as 1 is seldom possible due to the presence of signal nodes (not influenced by any other node in the network and therefore can retain any activity), negative feedback, and feed-forward loops. Therefore, we need to understand the upper limit of frustration allowed for a steady state of a given network.

For a state to be steady, the condition $\sum_j Adj_{ij}s_j(t)s_i(t) >= 0$ must be satisfied for all nodes in the network. Note that the magnitude of the term $Adj_{ij}s_i(t)s_j(t)$ is always 1. Therefore, for a node's activity to be retained, the number of frustrated incoming edges must always be less than or equal to the number of nonfrustrated incoming edges. Hence, frustration corresponding to a given node must be less than or equal to 0.5, a condition that directly extends to the frustration of a state.

## State strength

Given the strong similarity between influence matrix and the correlation matrix for all random networks, we asked whether influence matrix, which takes into account indirect interactions between nodes as well, is enough to explain the stability of the observed phenotypes. To answer this, we defined the strength of states – as a measure of how well a state is supported by the influence matrix – as follows:

$$Strength_S = \sum_{i,j=1}^{N} Infl_{ij}s_is_j; s_i, s_j \in \{0, 1\}$$

where $Infl_{ij}$ is the $(i, j)$th element of the influence matrix and $s_i$ and $s_j$ are the activities of the $i$th and $j$th node in the steady-state $S$ of interest. For WT EMP networks, the strength of terminal phenotypes is much higher (since all epithelial nodes have a positive influence on each other and so on) than the hybrid phenotypes (*Figure 4—figure supplement 2B and C*). If the influence matrix is sufficient to

explain the phenotypic stability, the correlation between strength and the stability metrics should be strongly positive for all networks (*Figure 4—figure supplement 2Di*). However, we find that the 57N 113E network that has the weakest teams among the WT EMP networks shows weak correlation between state strength and stability (*Figure 4—figure supplement 2Dii*). We calculated these correlation values for random networks and compared them against the corresponding team strengths (*Figure 4—figure supplement 2Diii*). Three out of five network sizes showed a positive correlation between the team strength and calculated correlations, indicating that the influence matrix can explain phenotypic stability better when team structure is strong.

At each parameter set, RACIPE integrates the model from multiple initial conditions and obtains steady states in the initial condition space. The output, hence, comprises of the collection of parameter sets and corresponding steady states obtained from the model. For the current analysis, we used a sample size of 10,000 for parameter sets and 100 for initial conditions. The parameters were sampled via a uniform distribution and the ODE integration was carried out using Euler's method of numerical integration.

