## [Editor Report]

This important article identifies topological metrics in gene regulatory networks that potentially predict the kinds of phenotypic steady states that the network allows. In particular, for epithelial–mesenchymal plasticity, the authors show compellingly that the relevant gene regulatory networks are structured as ‘teams’ that may be ‘strong,’ yielding stable phenotypes, or ‘weak,’ yielding unstable phenotypes prone to plasticity. The work would be of interest to researchers interested in systems biology and the nonlinear dynamics of biological systems, as well as biologists interested in gene regulatory networks and their (mis)functioning in cancer cells.

---

## [Decision Letter]

**Decision letter after peer review:**

Thank you for submitting your article "Landscape of epithelial mesenchymal plasticity as an emergent property of coordinated teams in regulatory networks" for consideration by *eLife*. Your article has been reviewed by 2 peer reviewers, and the evaluation has been overseen by a Reviewing Editor and Aleksandra Walczak as the Senior Editor. The following individual involved in the review of your submission has agreed to reveal their identity: Jean Clairambault (Reviewer #1).

Both reviewers found the results interesting and deserving of publication, but have also raised some concerns that require revisions in the manuscript. Most importantly:

1. The paper becomes very technical and will not be accessible to as general a readership as it deserves. To address this the reviewers have made some suggestions, such as defining terms and parameters more carefully and providing a better explanation of team identification in a general gene regulatory network. Please see the suggestions of Reviewer 2 in this regard. The authors should also consider moving some of the more technical discussion to supplementary material, and where they choose to keep the technical part in the main text they should try to explain more pedagogically the meaning of the equations and the connection to the biology.

2. It is not clear how generalizable the results are to other gene regulatory networks. Therefore, the authors should extend their studies to other networks, not just those relevant for epithelial-mesenchymal plasticity. For instance, they could examine small cell lung cancer and melanoma networks. In addition, the authors should try to comment, even if speculatively, on the implications of their results for the interplay between cooperation and competition in cellular populations both in unstructured colonies as well as multicellular organisms (please see the comments of Reviewer 1).

*Reviewer #1 (Recommendations for the authors):*

The authors here – and this is absolutely normal – focus on their main usual topic, EMP, which has been reported to be physiologically present in development and wound healing (going as far as limb regeneration in axolotl), but is most often studied in cancer. Could the authors sketch possible studies in other fields in which tissue indetermination and plasticity have been reported, e.g., cases of de-differentiation or transdifferentiation in cancer?

Other (teleologically conceptual) words for coordination are compatibility and cooperativity, two features that are inherent to cohesive multicellularity, yielding effective division of labour in a healthy organism. In this respect and in their present study, the authors have shown that organisation of GRNs in 'strong' teams contributes to the stability of terminal phenotypes. This may account for the separation of phenotypes in different tissues. However, the nature of compatibility within GRNs in cells of a given tissue seems to be absent from their analyses. Similarly, the intensity of links between teams of GRNs in different tissues, producing cooperation, would be another step forward to study the cohesion of a multicellular organism, which must reject plasticity in general, and MEP in particular. Could the authors suggest extensions of their well-designed methodology to study such aspects of cohesion in multicellular organisms?

Other teams of researchers have studied differentiation from the point of view of the emergence of multicellularity, which is a fundamental question in developmental biology, with solutions that may be recapitulated by cancer cell populations showing EMP. In particular, Kunihiko Kaneko's lab in Tokyo (https://doi.org/10.7551/mitpress/10525.003 of 2016) has developed a conceptual representation of the stability of phenotypes related to stationarity of expression of genes in collections, and instability related to oscillatory solutions, that might be made close to the present manuscript. Of course, their interest is more devoted to induced pluripotent stem cells (iPSCs) and the Yamanaka genes *Sox2*, Klf4, Oct4, and c-Myc (see arXiv:2109.04739v2 of 2021); however, the idea of 'strong' and 'weak' teams of GRNs might also be investigated from this point of view. Could the authors comment on this suggestion?

*Reviewer #2 (Recommendations for the authors):*

This paper has great theoretical potential in the context of Systems Biology. However, it becomes too technical too quickly, and some mathematical concepts are not properly defined. This might "scare" some readers away and makes it difficult to identify the most important take-home messages in each section. I hope that with the more detailed comments below, I can help the authors improve the quality of their manuscript so that it will get the full deserved attention from any reader willing to learn more about Systems Biology. The authors could consider also shortening the paper.

– In the abstract, the authors talk about SCLC without previous definition. In the first sentence, they use the concept of coordinated "teams" of nodes, but nodes per se are never defined.

– What is the biological difference between the 5 networks used to describe epithelial-mesenchymal plasticity?

– It might be that the legend of the color bar in the adjacent matrix in Figure 1B (left) is wrong, and the authors are displaying the "adjacency" value. From the plot, I understand that this parameter can take only the values +1,-1 and 0. However, I did not find it defined anywhere in the main text. In addition, it would be helpful to indicate what is the difference between the nodes annotated in the y tick (input node of an edge?) and the x tick (output node of an edge?).

– When the authors introduce the equation for the "influence" in Figure 1B (left), most parameters are left undefined (by the end of reading the manuscript I found some description in section 4, however, I did not manage to understand the calculation). Is not Adj_{max}^{l} always equal to 1? How is Adj^{l} (or J^l according to Equation (5)) calculated? Are all the possible paths connecting two nodes considered in the Equation? It would make sense to refer to section 4 when the equation for "influence" appears for the first time and spend more text making all the proper definitions.

– Why do VIM, TCF3 and KLF8 not cluster with FOXC2 etc, and why does CDH1 not cluster with miR101 et al. according to the Influence profile in Figure 1B (right) (or according to correlation in Figure 1D,1E)? Do I read it correctly when I say that CDH1 is not influencing any other node in the network? Is its value then important to define phenotypes? Can we accordingly remove this node from the network without any consequence of the phenotype frequency? Can this be used as an approach to simplify network topology?

– How is the number of node groups identified in the different WT and randomized networks? Is it always equal to 3 (input/output, and 2 types of core nodes)? I imagine this to be a crucial aspect of the calculation of the group strength (Figure 1B, right). In addition, what is the parameter n_{II} in Figure 1B, bottom-right?

– I think that in the last paragraph of section 2.1, the authors need to cite together Figure 1D and 1E, and where they cite Figure 1E they refer to Figure 1F.

– In section 2.2, the authors state that the presence of two strong distinct teams can contribute to bimodal distributions in SSF plots. Based on the observations obtained from Chauhan et al., 2021, there two strong teams give rise to four strong stable phenotypes. Therefore, I wonder how generalizable the statement is. Can the authors comment on this?

– In the 3rd paragraph of section 2.2, the authors refer to the Methods section while describing the concept of frustration. However, I could not find any explanation of frustration in the Methods.

– I do not understand how much the analysis of frustration values is adding to the steady-state frequency. Intuitively, these two concepts are intimately related: less stable steady-state has higher frustration. The authors could consider removing one of the two to simplify the paper. From my point of view, SSF is a better measurement, since the authors can see that low frustration values can have practically any SSF while high frustration values only have low SSF (Figure 2D, 2F). Therefore, it seems to me that SSF provides more information.

– In Figure 2B, the authors again provide some equations that lack definitions. What are W_{ij} and s_i_? What do "ON" and "OFF" mean?

– In Figure 2B, I would use SSF consistently when indicating steady-state frequency values. Is Frequency in panel D the same steady-state frequency as in panel A?

– How do the other networks look like when plotting "Frustration" versus "SFF" (Figure 2F)?

– In Figure 3A, it would be helpful to explain what read and blue means, and what would be the value of coherence in this particular sketch.

– I find it surprising that some steady states have a very high coherence and a very low frequency (Figure 3E). Can the authors comment on that?

– Figure 4Aii (S5B) is very important: it shows for the first time that the mean group strength value might indeed be related to the bimodality of steady-state types. However, as it is right now, the distribution of frustration+coherence values for the high mean group strength networks is not seen. I suggest the authors show the distributions of frustration, coherence, and frustration+coherence values for high and low mean group strength networks in different panels, and then they explain how are they obtaining the composite (maybe this magnitude should be introduced before and merge Figure 2 and 3 into one). I find the histograms of frustration+coherence for 57N113E, 20N40E and 26N100E provided in Figure S5B doubtful.

– The equation of strength of a state, in Figure 4B, should be better explained. What is s_i_?

– Are the capital S in Equation (8-10) the same as the small s in Equation 7?

– Network 57N113E is persistently giving less clear results. Can the authors identify whether the way the network was constructed has some defect? In other words, could their theoretical approach help validate experimentally determined gene regulatory networks?

---

## [Author Response]

1. The paper becomes very technical and will not be accessible to as general a readership as it deserves. To address this the reviewers have made some suggestions, such as defining terms and parameters more carefully and providing a better explanation of team identification in a general gene regulatory network. Please see the suggestions of Reviewer 2 in this regard. The authors should also consider moving some of the more technical discussion to supplementary material, and where they choose to keep the technical part in the main text they should try to explain more pedagogically the meaning of the equations and the connection to the biology.2. It is not clear how generalizable the results are to other gene regulatory networks. Therefore, the authors should extend their studies to other networks, not just those relevant for epithelial-mesenchymal plasticity. For instance, they could examine small cell lung cancer and melanoma networks. In addition, the authors should try to comment, even if speculatively, on the implications of their results for the interplay between cooperation and competition in cellular populations both in unstructured colonies as well as multicellular organisms (please see the comments of Reviewer 1).

We are grateful to you for providing us an opportunity to revise our manuscript entitled “Landscape of epithelial mesenchymal plasticity as an emergent property of coordinated teams in regulatory networks”. In the revised manuscript, we have carefully considered reviewers’ comments and suggestions. As instructed, we have attempted to succinctly explain changes made in reaction to all comments. We have replied to each comment in point-by-point fashion, and highlighted the changes made in the manuscript. The reviewers’ comments were very helpful overall, and we are appreciative of such constructive feedback on our original submission.

Specifically, we have made these two major changes in our manuscript:

1. We recognize the issue raised by the reviewer that the previous version of our manuscript became too technical too soon. To address this limitation, we have accordingly re-written a significant portion of the manuscript, with less technical details and more interpretations and pedagogical explanations. To improve the accessibility, we also rewrote the methods section with notations and figure-wise methods described.

2. We explore the generalizability of the results by including the analysis for additional gene regulatory networks not implicated in epithelial-mesenchymal plasticity, but those regulating phenotypic heterogeneity in small cell lung cancer (SCLC) and in Melanoma (Figure 9, Figure 9 —figure supplement 1), as suggested. We have also included discussion on how the idea of teams can be extended to understand differentiation and multicellularity.

Reviewer #1 (Recommendations for the authors):The authors here – and this is absolutely normal – focus on their main usual topic, EMP, which has been reported to be physiologically present in development and wound healing (going as far as limb regeneration in axolotl), but is most often studied in cancer. Could the authors sketch possible studies in other fields in which tissue indetermination and plasticity have been reported, e.g., cases of de-differentiation or transdifferentiation in cancer?

We thank the reviewer for the suggestion. We have now included an analysis for GRNs in the context of small cell lung cancer and melanoma (Figure 9, Figure 9 —figure supplement 1) which also show the formation of “teams” and a limited phenotypic repertoire, despite their complexity.

We have also included a brief discussion on how understanding the presence and role of such teams can help connect various axes of plasticity that can contribute to cancer cell survival:

“Recent studies have started to identify the connections between different axes of plasticity underlying cancer cells, such as EMP, drug resistance, immune evasion, and dormancy. While, in most cases, these connections have been explored using small-scale networks, the presence of teams provides an intuitive way of reducing the complexity of large networks and therefore enables the coordination of multiple axes of plasticity at a large scale; for example, EMP and drug resistance (Sahoo et al., 2021).”

Other (teleologically conceptual) words for coordination are compatibility and cooperativity, two features that are inherent to cohesive multicellularity, yielding effective division of labour in a healthy organism. In this respect and in their present study, the authors have shown that organisation of GRNs in 'strong' teams contributes to the stability of terminal phenotypes. This may account for the separation of phenotypes in different tissues. However, the nature of compatibility within GRNs in cells of a given tissue seems to be absent from their analyses. Similarly, the intensity of links between teams of GRNs in different tissues, producing cooperation, would be another step forward to study the cohesion of a multicellular organism, which must reject plasticity in general, and MEP in particular. Could the authors suggest extensions of their well-designed methodology to study such aspects of cohesion in multicellular organisms?

We are thankful to the reviewer for their suggestion. We appreciate that the ideas proposed by the reviewer are indeed a natural and important extension of the ideas investigated in this manuscript. We have included a detailed discussion on using the idea of teams to understand/model the evolution of multicellularity. We find that the formalism itself needs to evolve to capture these processes, and have proposed a few ways in which this can be possibly attempted:

“Here we see teams in terms of intracellular regulatory networks. However, this framework of identifying the composition of two (or more) teams acting together to reinforce each other in scenarios of competing outcomes can be applied more broadly, particularly in multicellularity. […] This interpretation does assume that the team acts as a single component, which has to be validated for different biochemical and spatial interactions of cells within a team.”

Other teams of researchers have studied differentiation from the point of view of the emergence of multicellularity, which is a fundamental question in developmental biology, with solutions that may be recapitulated by cancer cell populations showing EMP. In particular, Kunihiko Kaneko's lab in Tokyo (https://doi.org/10.7551/mitpress/10525.003 of 2016) has developed a conceptual representation of the stability of phenotypes related to stationarity of expression of genes in collections, and instability related to oscillatory solutions, that might be made close to the present manuscript. Of course, their interest is more devoted to induced pluripotent stem cells (iPSCs) and the Yamanaka genes Sox2, Klf4, Oct4, and c-Myc (see arXiv:2109.04739v2 of 2021); however, the idea of 'strong' and 'weak' teams of GRNs might also be investigated from this point of view. Could the authors comment on this suggestion?

We thank the reviewer for the very constructive feedback. We have discussed the implications of teams in sequential differentiation.

“Teams of nodes forming a toggle switch can be an excellent way to explain the canalization property observed in development. […] Identifying changes in regulatory networks that would be required to implement these rearrangements will be an exciting direction.”

Reviewer #2 (Recommendations for the authors):This paper has great theoretical potential in the context of Systems Biology. However, it becomes too technical too quickly, and some mathematical concepts are not properly defined. This might "scare" some readers away and makes it difficult to identify the most important take-home messages in each section. I hope that with the more detailed comments below, I can help the authors improve the quality of their manuscript so that it will get the full deserved attention from any reader willing to learn more about Systems Biology. The authors could consider also shortening the paper.– In the abstract, the authors talk about SCLC without previous definition. In the first sentence, they use the concept of coordinated "teams" of nodes, but nodes per se are never defined.

We thank the reviewer for the suggestion. We have now updated the abstract to improve clarity, including the details of constitution of “teams” of nodes, as given below:

“Elucidating the design principles of regulatory networks driving cellular decision-making has fundamental implications in mapping and eventually controlling cell-fate decisions. […] We propose “teams” of nodes as a network design principle that can drive cell-fate canalization in diverse cellular decision-making processes.”

– What is the biological difference between the 5 networks used to describe epithelial-mesenchymal plasticity?

These networks differ in the biological context in which they were identified. For instance, 57N113E (Font-Clos et al. 2018) includes factors driving Mesenchymal-Epithelial Transition (MET) in an earlier network constructed for hepatocellular carcinoma related EMT (Steinway et al. Cancer Res 2015). However, as far as our study is concerned, we focus on topological differences, mainly in terms of network size and density, among them, as stated now in in the first section of the results.

“We chose these networks to range over various sizes and densities (18N 33E to 57N 113E, where N is the number of nodes and E is the number of edges in the network). Each of these networks depicts the regulation of EMP at the transcriptional/post-transcriptional level (compiled under different biological/experimental contexts). Therefore, each node is either a transcription factor or a microRNA, and each edge represents transcriptional or post-transcriptional activation or inhibition.”

– It might be that the legend of the color bar in the adjacent matrix in Figure 1B (left) is wrong, and the authors are displaying the "adjacency" value. From the plot, I understand that this parameter can take only the values +1,-1 and 0. However, I did not find it defined anywhere in the main text. In addition, it would be helpful to indicate what is the difference between the nodes annotated in the y tick (input node of an edge?) and the x tick (output node of an edge?).

We thank the reviewer for this comment. We have updated the legend of the color bar (earlier Figure 1B, which is now Figure 4A). We have added a description of the x and y axis in the legend.

“Adjacency matrix of the 22N 82E network. Each row depicts the links originating from the node (i.e., input) corresponding to the row (y-axis) and all other nodes (x-axis, outputs). The color represents the nature of the edge: red for activating links, blue for inhibiting links, and white for no links. The formula for the conversion of adjacency matrix to influence matrix is given below the panel.”

We have also updated the description of the adjacency values in methods section 4.1.

– When the authors introduce the equation for the "influence" in Figure 1B (left), most parameters are left undefined (by the end of reading the manuscript I found some description in section 4, however, I did not manage to understand the calculation). Is not Adj_{max}^{l} always equal to 1? How is Adj^{l} (or J^l according to Equation (5)) calculated? Are all the possible paths connecting two nodes considered in the Equation? It would make sense to refer to section 4 when the equation for "influence" appears for the first time and spend more text making all the proper definitions.

We have updated the text with the reference to the methods section 4.6.1 and have updated the description of influence matrix calculation for a clearer understanding.

“The Influence Matrix, as the name suggests, is a matrix where each element at (i,j) position records the influence of i^th^ node on the j^th^ node in the network, mediated through one or more serially connected edges that form a path from ith node to the j^th^ node in the network. […] The division with lmax ensures that the elements of Inflmax are constrained between -1 and 1.”

The Adj^l^ and Adjmax^l^ represent the l^th^ power of the matrices Adj and Adjmax respectively. The values in Adjmax represent the magnitude of the interaction between two nodes, without the sign. The matrix power, as the reviewer rightly understood, takes into account all paths of length l between any two nodes and in a given direction. The maximum value that this path-sum can have for the given network structure is represented by Adjmax.

– Why do VIM, TCF3 and KLF8 not cluster with FOXC2 etc, and why does CDH1 not cluster with miR101 et al. according to the Influence profile in Figure 1B (right) (or according to correlation in Figure 1D,1E)? Do I read it correctly when I say that CDH1 is not influencing any other node in the network? Is its value then important to define phenotypes? Can we accordingly remove this node from the network without any consequence of the phenotype frequency? Can this be used as an approach to simplify network topology?

We agree with the reviewer on this point. The output nodes VIM, TCF3, CDH1 etc. do not influence any other nodes in the network, hence their contribution to dynamics is negligible. We have now clarified the three types of nodes in any network:

“We classify these nodes into two categories based on their topological configuration: “peripheral” nodes and “core” nodes. […] Similar classification of nodes has been implemented for other networks (Figure 1 —figure supplement 1A, Silveira and Mombach (2019), Silveira et al. (2020), Huang et al. (2017), Tripathi et al. (2020a), Font-Clos et al. (2018)).”

In single-node coherence calculations (Figure 6A, Figure 6 —figure supplement 1A), we found that all states show a coherence of 1 upon perturbing the output node, indicating the lack of effect of perturbation of output nodes on the consequent dynamics. More importantly, we find that the signal nodes also do not affect the dynamics much, most likely due to reinforcement present from other nodes due to direct/indirect paths. To establish this effect further, we calculated the coherence specific to the core nodes when the signal nodes are perturbed (Figure 6 —figure supplement 2). Perturbing signal nodes does not change the configuration of the core nodes in terminal phenotypes. We agree with the reviewer that eliminating such nodes can be a way to simplify network topology.

– How is the number of node groups identified in the different WT and randomized networks? Is it always equal to 3 (input/output, and 2 types of core nodes)? I imagine this to be a crucial aspect of the calculation of the group strength (Figure 1B, right). In addition, what is the parameter n_{II} in Figure 1B, bottom-right?

We thank the reviewer for pointing out this aspect. We have now clearly explained this in the results and the methods. We forced the clustering algorithm to identify only two teams in these networks, in compliance with the observation of two teams in the WT EMP networks.

“3. Apply hierarchical clustering (hclust in R 4.1.2) on the (influence) matrix

4. Break the resultant dendrogram in two (cutree in R 4.1.2), resulting in the two teams of nodes.”

We have also updated the formulae and the figure caption for the influence matrix (Figure 4A, 4B, previously Figure 1B) with a detailed explanation of various terms involved in the formulae. The details are also provided in the corresponding methods sections.

– I think that in the last paragraph of section 2.1, the authors need to cite together Figure 1D and 1E, and where they cite Figure 1E they refer to Figure 1F.

We thank the reviewer for identifying this mistake. We have now updated the text with proper citations.

– In section 2.2, the authors state that the presence of two strong distinct teams can contribute to bimodal distributions in SSF plots. Based on the observations obtained from Chauhan et al., 2021, there two strong teams give rise to four strong stable phenotypes. Therefore, I wonder how generalizable the statement is. Can the authors comment on this?

In SCLC, there are two groups of states, one with relatively very high SSF (the four steady states) and another with relatively very low SSF (6 steady states), leading to the bimodality in the distribution of SSF. Such bimodality is also seen in five different EMP networks (Figure 2A).

Furthermore, the four steady states can be classified into two sets of two steady states, with the only difference between the states in a set being the activity of NEUROD1. This classification is consistent with our categorization of EMP steady states into Epithelial and Mesenchymal sets of steady states (or phenotypes) that have the degeneracy in terms of activities of the signal nodes (miR205, miR30c and miR9 in 22N 82E network), but show identical activity for the corresponding core nodes. We have performed this analysis on the melanoma GRN as well (Figure 9 in this paper, and previous analysis in Pillai and Jolly, iScience 2021) and found that the two teams give rise to two sets of highly stable steady states (i.e., two strong phenotypes): proliferative and invasive.

– In the 3rd paragraph of section 2.2, the authors refer to the Methods section while describing the concept of frustration. However, I could not find any explanation of frustration in the Methods.

We thank the reviewer for pointing this out. We have added a section describing frustration in the methods now.

– I do not understand how much the analysis of frustration values is adding to the steady-state frequency. Intuitively, these two concepts are intimately related: less stable steady-state has higher frustration. The authors could consider removing one of the two to simplify the paper. From my point of view, SSF is a better measurement, since the authors can see that low frustration values can have practically any SSF while high frustration values only have low SSF (Figure 2D, 2F). Therefore, it seems to me that SSF provides more information.

We are grateful for this insight by the reviewer. We included frustration as a stability metric here for two reasons.

First, we observe that the correlation between frustration and SSF (or coherence) is weakened for random networks and is stronger only when teams are stronger (Figure 3 —figure supplement 1D). We interpret frustration as a measure of how a network supports the steady states and suggest that support by the network topology alone is not always enough to explain phenotypic stability, but this relationship improves with strong teams, indicating the importance of the “teams” structure in maintaining the relationship between the different stability measures.

Second, various biological networks have been proposed to have lower frustration than most of their random counterparts (Tripathi *et al.* Phys Rev Letter 2020). Our results here argue that strong “teams” in biological networks can be an important mechanism underlying the observed minimal frustration in many biological networks.

– In Figure 2B, the authors again provide some equations that lack definitions. What are W_{ij} and s_i_? What do "ON" and "OFF" mean?

We have updated Figure 3A (previously Figure 2B) with the more consistent parameter Adj_ij_, which represents the interaction from ith node to jth node. We have also included descriptions of s_i_ in the corresponding caption:

“Adj_ij represents the interaction from ith node to jth node, s_i_ and s_j_ represent the activity of the ith node and the jth node for a given state S”

– In Figure 2B, I would use SSF consistently when indicating steady-state frequency values. Is Frequency in panel D the same steady-state frequency as in panel A?

We have updated the usage of SSF across the manuscript.

– How do the other networks look like when plotting "Frustration" versus "SFF" (Figure 2F)?

In the revised manuscript, Figure 2F (“frustration versus SSF”) is now Figure 3B. We have also included the correlation for these metrics for corresponding random networks in Figure 3 —figure supplement 1B.

– In Figure 3A, it would be helpful to explain what read and blue means, and what would be the value of coherence in this particular sketch.

We thank the reviewer for this suggestion. We have added the value of coherence in the depiction (now Figure 2B) and provided explanations for different parts of the figure in the caption:

“The blue balls indicate unperturbed steady state (P1, say). The green and dark blue balls represent the perturbations given to the steady state. The red balls represent a different steady state that the system reached after the perturbation. The fraction of perturbations that reverted to the original state P1 (3 out of 7 balls) is calculated as coherence.”

– I find it surprising that some steady states have a very high coherence and a very low frequency (Figure 3E). Can the authors comment on that?

We apologise for the apparent confusion. SSF is a global stability metric but coherence is a local stability measure. Thus, with more number of steady states, the maximum SSF value will decrease (which is what happens in 57N 113E). Thus, while the absolute values of coherence and SSF may vary, depending on the network, the correlation between them remains positive, even for random networks (Figure S4C). We have also included this aspect briefly in the main text, referring to the different range of values noted for coherence vs. those for SSF, as mentioned below:

“Coherence being a local stability measure, is less dependent on the other steady states of the network. Therefore, the absolute coherence values can be compared across networks, unlike SSF (range of y axis values in Figure 2A for SSF vs 2C for coherence).”

– Figure 4Aii (S5B) is very important: it shows for the first time that the mean group strength value might indeed be related to the bimodality of steady-state types. However, as it is right now, the distribution of frustration+coherence values for the high mean group strength networks is not seen. I suggest the authors show the distributions of frustration, coherence, and frustration+coherence values for high and low mean group strength networks in different panels, and then they explain how are they obtaining the composite (maybe this magnitude should be introduced before and merge Figure 2 and 3 into one). I find the histograms of frustration+coherence for 57N113E, 20N40E and 26N100E provided in Figure S5B doubtful.

We agree with the reviewer on the confusion this figure had caused. We have now removed the frustration + coherence density plots (Figure 5). We realized that the distinction between terminal and hybrid phenotypes can be visualized without this composite axis as well.

We have instead characterized the effect of team strength on bimodality through violin plots of bimodality coefficient for SSF and Coherence against team strength (Figure 5B, iii; Figure 5 —figure supplement 1C and Figure 5 —figure supplement 2C). As for the networks 57N113E, 20N40E and 26N100E, we have highlighted the difference in the observations made in the scatterplots (Figure 5C, Figure 5 —figure supplement 1D) and have explained the reasons for it:

“To better visualize the effect of bimodality coefficient, we took 10 random networks each with the highest and lowest team strengths, and mapped the frustration and coherence of their steady states (Figure 5C, Figure 5 —figure supplement 1D). For networks with high Ts (red points), we clearly see two groups of steady states based on the relative stability (high coherence – low frustration and low coherence – high frustration). While such distinction of two groups of steady states is lost in random networks of low Ts corresponding to 22N 82E, 18N 33E and 20N 40E, it was maintained in 26N 100E and 57N 113E random networks. This observation strengthens the trend that high team strength corresponds to a bimodal landscape, while at low team strength, bimodality of the phenotypic stability landscape remains unpredictable.”

– The equation of strength of a state, in Figure 4B, should be better explained. What is s_i_?

We have included more details about the state strength but decided to move the description as well as the figures to supplementary to reduce the confusion in the narrative:Strengths=∑i,j=1NInflijsisj;si,sjϵ{0,1}“Where Infl_ij is the (i, j)th element of the Influence matrix and s_i_ and s_j_ are the activities of the ith and jth node in the steady state S of interest.”

– Are the capital S in Equation (8-10) the same as the small s in Equation 7?

We have now made the notation consistent and added a glossary of the notations at the beginning of the methods section. S represents a steady state, while s represents the activity of individual nodes in a given state S.

– Network 57N113E is persistently giving less clear results. Can the authors identify whether the way the network was constructed has some defect? In other words, could their theoretical approach help validate experimentally determined gene regulatory networks?

A possible reason we can postulate why 57N 113E gives less clear results is that it has weaker team strength as compared to the other four networks investigated. Through our analysis of random networks (of varying team strength) corresponding to various WT networks, we noticed that many static and dynamic metrics do not show as clear trends when the team strength is weaker. For instance, maximum SSF, maximum coherence and coherence bimodality coefficient (Figure 5B, i-iii) are consistently higher for networks with strong teams; however, at weak team strength, the variability in these values is quite high, thus weakening the trends. These results suggest that the association of maximum SSF, maximum coherence, minimum frustration with team strength is more pronounced at larger team strength, and thus the weak team strength in 57N 113E network weakened its trends.